



# The 2018 Lake Muir earthquake sequence, southwest Western Australia: rethinking Australian stable continental region earthquakes

Dan J. Clark[1], Sarah Brennand[1], Gregory Brenn[1], Trevor I. Allen[1], Matthew C. Garthwaite[1], Sean Standen[2]

[1] Positioning and Community Safety Division, Geoscience Australia, GPO Box 378 Canberra ACT, Australia
[2] The University of Western Australia, 35 Stirling Hwy, Crawley, Western Australia, Australia

*Correspondence to*: Dan J. Clark (dan.clark@ga.gov.au)

**Abstract.** Modern geodetic and seismic monitoring tools are enabling the study of moderate-sized earthquake sequences in unprecedented detail. Here we use a variety of methods to examine surface deformation caused by a sequence of earthquakes near Lake Muir in southwest Western Australia in 2018. A shallow MW 5.3 earthquake on the 16th of September 2018 was followed on the 8th of November 2018 by a MW 5.2 event in the same region. Focal mechanisms for the events suggest reverse and strike-slip rupture, respectively. Interferometric Synthetic Aperture Radar (InSAR) analysis of the events suggests that the ruptures are in part spatially coincident. Field mapping, guided by the InSAR results, reveals that the first event produced an approximately 3 km long and up to 0.5 m high west-facing surface rupture, consistent with slip on a moderately east-dipping fault. Double difference hypocentre relocation of aftershocks using data from rapidly deployed seismic instrumentation confirms an east-dipping rupture plane for the first event, and shows a concentration located at the northern end of the rupture where the InSAR suggests greatest vertical displacement. The November event resulted from rupture on a northeast-trending strike-slip fault. UAV-derived digital terrain models (differenced with pre-event LiDAR) reveal a surface deformation envelope consistent with the InSAR for the first event, but could not discern deformation unique to the second event. New rupture length versus magnitude scaling relationships developed for non-extended cratonic regions as part of this study allow for the distinction between "visible" surface rupture lengths (VSRL) from field-mapping and "detectable" surface rupture lengths (DSRL) from remote sensing techniques such as InSAR, and suggest longer ruptures for a given magnitude than implied by commonly used scaling relationships.





## 1 Introduction

In Australia, nine earthquakes are known to have produced surface rupture in historical times (Table 1, Figure 1). These ruptures are located exclusively in the Precambrian cratonic crust of central and western Australia (Clark et al., 2014a), and are associated with moment magnitude ($M_W$) events in the range of 4.73 - 6.76 (Clark et al., 2014b;Clark and Allen, 2018). The 2007 Katanning $M_W$ 4.73 event was the first earthquake in Australia where the surface deformation field was imaged by InSAR (Dawson et al., 2008), a technique which enables the mapping in unprecedented detail of surface deformation relating to moderate to small, shallow earthquakes (Dawson and Tregoning, 2007); much greater than could be mapped on the ground using traditional paleoseismological means. The InSAR results for the Katanning event, derived from Japanese ALOS-PALSAR data, showed surface deformation along a linear front ~1.26 km in length, of which 0.2 km was recognisable on the ground as a scarp up to 0.1 m high (Vic Dent, pers. comm. 2007). This finding dramatically reduced the expected threshold magnitude for surface rupture in cratonic Australia, and offered a means of gaining a more comprehensive understanding of the structural and seismotectonic context, and the landscape effects, of the remarkably shallow intraplate ruptures experienced in cratonic Australia. In this paper we report on the geological setting and characteristics of the surface deformation relating to two $M_W$ >5.0 earthquakes that occurred in 2018 near Lake Muir in the southwest of Western Australia, as imaged using Sentinel-1 InSAR data. The InSAR observations of surface deformation were validated using field observations, unmanned aerial vehicle (UAV)-acquired high resolution digital elevation data, and hypocentres calculated from a local seismic network deployed in the days following the first, $M_W$ 5.3, event.

### 1.1 The Lake Muir Earthquake Sequence

The $M_L$ 5.7 ($M_W$ 5.3) Lake Muir earthquake occurred at 04:56:24 (UTC) on 16[th] September 2018 in a rural area of southwest Western Australia, ~60 km east of the town of Manjimup, and a similar distance north of the town of Walpole (Figure 2). Approximately 20 km from the epicentre, relatively modest Modified Mercalli Intensity (MMI) values of VI were reported (Allen et al., 2019). With the exception of one unoccupied dwelling in the immediate epicentral area, which was extensively damaged, only minor damage and no injuries were reported. The event was widely felt throughout the Perth Basin, including the Perth metropolitan region, over 300 km away. Focal mechanisms suggest a reverse faulting mechanism, with a minor dextral transcurrent component, and moderately east and northwest dipping nodal planes (Figure 2, Table 2). Geoscience Australia recorded a magnitude $M_L$ 3.4 foreshock three days prior to the main shock. A protracted aftershock sequence, comprising hundreds of located events, was punctuated by a $M_L$ 4.7 event on 12[th] October at 16:31:30 (UTC) (Figure 3, Table 2). Almost two months after the 16[th] September $M_W$ 5.3 event, a $M_L$ 5.4 ($M_W$ 5.2) event occurred on the 8[th] November at 21:07:01 (UTC). Within the error estimates of the hypocentral determinations, this event was co-located with the 16[th] September event (Table 2). The focal mechanisms calculated for the November event indicate predominantly dextral strike-slip faulting, with steeply northwest and southwest dipping nodal planes. The percentage double-couple from the USGS W-phase moment tensor solution is 31% (Figure 2, Table 2). This event was felt much more strongly and widely than the slightly



larger first event, with MMI of VII to VIII being recorded close to the epicentre. There are several factors that might explain the relatively greater number (and density) of felt reports from the 8th November earthquake, including: 1) the time of day, which was early morning, when people are more likely to be stationary and thus more likely to perceive ground shaking; 2) differences in the ground motion radiation pattern and/or stress drop of the earthquake that may have yielded greater shaking in population centres at perceptible frequencies; and/or 3) greater community awareness of the earthquake sequence and where to find sources of further information. Surface rupture relating to the September event was initially identified with InSAR, and has been subsequently verified by ground survey (this paper).

**1.2 Geological and seismotectonic setting of the Lake Muir earthquake sequence**

The Lake Muir earthquake sequence occurred within 1700-1600 Ma rocks of the Biranup Zone of the Proterozoic Albany Fraser Orogen (Spaggiari et al., 2009;Fitzsimons and Buchan, 2005) (Figure 2). During the Mesoproterozoic Albany Fraser orogeny these rocks were thrust to the north over the 3000-2600 Ma rocks of the Northern Foreland of the Albany Fraser Orogen (the portion of the Archaean Yilgarn Craton that was reworked during the Albany–Fraser Orogeny) along moderately south-dipping faults. In the Lake Muir region, the dominant east to east-southeast striking structural grain is associated with the major structures bounding these thrust sheets. East-striking structural and lithological trends relating to the Albany Fraser Orogen are cut by northwest trending structures related to Proterozoic to earliest Phanerozoic movement on the Darling Fault Zone to the west (Janssen et al., 2003). The Boyagin dyke swarm cuts across much of the southwest, including the Yilgarn Craton and Albany–Fraser Orogen (Prider, 1948; Myers, 1990a; Harris and Li, 1995), and is subparallel to this structural trend. Minor north- and northeast-trending structures may relate to Gondwana breakup or later movement focused on the Darling Fault, parallel structures to the west thereof (e.g., the Dunsborough and Busselton faults), and associated oblique structures (Harris, 1996;Olierook et al., 2015).

The Lake Muir region lies near to the southern boundary of a broad band of relatively high seismicity crossing the southwest corner of Western Australia known as the Southwest Seismic Zone (SWSZ, Doyle, 1971), which is one of the most seismically active regions in Australia (e.g. Leonard, 2008;Allen et al., 2018). Earthquake activity in the SWSZ appears to have increased significantly since the 1940s (Leonard, 2008), and it has generated five of the nine known Australian historic surface ruptures (Table 1). In addition to large shallow events and scattered smaller events, the SWSZ has produced several dozen earthquake swarms in the last 40 years, including the Burakin swarm of 2001-2002 during which ~18,000 events (including three above $M_L$ 5.0) were recorded in a period of only a few months (Leonard, 2002;Dent, 2016;Allen et al., 2006). While most swarm centres occur within the SWSZ, they have a broader distribution across the southwest of Western Australia (Dent, 2016), a pattern that is similar to fault scarps relating to pre-historic events (Clark, 2010;Clark et al., 2012). The relatively uniform spatial distribution of north-trending reverse fault scarps is consistent with scarp formation under conditions imposed by the contemporary east-west oriented compressive crustal stress regime (e.g. Rajabi et al., 2017), and suggests that strain is uniformly distributed over the Yilgarn Craton over the timescale recorded in the land surface (*ca.* 100 kyr or more) (Leonard and Clark, 2011).



### 1.3 Landscape context of the Lake Muir earthquake sequence

Interrogation of a 2012 LiDAR dataset (see Supplementary Information) revealed the presence of grain in the landscape that mimics structural trends evident in the continental-scale magnetic data (cf. Figures 2 and 3). The main ridge-line, valley and drainage trends in the study area are broadly to the east-southeast and north, with a subordinate northwest trend (black double-ended arrows in Figure 3). Ridges are subdued, broad and undulating. Valley floors are flat-bottomed, and are locally occupied by lakes and swamps, implying the presence of significant alluvial or sedimentary infill (cf. Commander et al., 2001;Smith, 2010). Lunettes are developed on the east and southeast sides of most lakes, often with evidence for minor remobilisation into parabolic dunes. Knick-point retreat from a regional topographic low to the southwest occupied by Lake Muir is observed in several instances to be rejuvenating the drainage within flat-bottomed valleys, resulting in the removal of valley-fill sediments and the accentuation of structural trends evident in the alignment of adjacent linear ridges. There is no obvious landscape evidence for sharp vertical dislocations of valley floor sediments that might relate to Late Pleistocene or Holocene surface-rupturing seismic events.

### 2 Methods

### 2.1 InSAR processing method

For each of the two earthquake events we generated a coseismic interferogram from two Sentinel-1 interferometric wide swath SAR images (earthquake 1: 14th and 26th September 2018; earthquake 2: 1st and 13th November 2018) using a standard interferometric SAR processing workflow implemented with the Gamma software (Wegmüller and Werner, 1997). The topographic component of the phase signal was removed using a 1-arc second SRTM digital elevation model (Geoscience Australia, 2011) and the orbital component was removed using Precise Orbital Ephemerides (POD) products. Each interferogram was sub-sampled (multi-looked) eight times in range and two times in azimuth to reduce phase noise. An adaptive spatial filter (Goldstein and Werner, 1998) with exponent 0.5 was applied to each interferogram to further reduce phase noise prior to automatic unwrapping using a minimum cost flow algorithm (Costantini, 1998). The automatically unwrapped interferogram for the September earthquake had significant errors due to poorer phase coherence around the surface rupture zone. A manual approach to unwrapping this interferogram was therefore applied using the branch-cut method (Goldstein et al., 1988).

### 2.2 Field observation and UAV digital terrain models

The epicentral region of the earthquakes were visited over three days (12-14th November 2018), less than a week after the November event. The scarp produced by the September event was mapped, and evidence for rejuvenation of the scarp during the November event was assessed. A hand trench was excavated across the central section of the September scarp to assess fault dip and displacement.



In order to recover the surface deformation envelope associated with the events, aerial photographs were acquired with a DJI Phantom 4 UAV in an approximately 500 m-wide swath along a 2 km length of the September event scarp. An approximately 2 km-long cross-line was also flown, extending eastward from the scarp across the region of most significant surface deformation indicated in the InSAR imagery (Figure 3). Ground control was provided by an RTK GPS, with mean horizontal and vertical measurement uncertainties of 0.026±0.025 m and 0.056±0.055 m, respectively. A third mission covered the

southern extent of surface deformation indicated in the InSAR imagery (Figure 3). The results of this mission are presented in the Supplementary Information.

The image dataset was processed using a Structure-from-Motion (SfM) and multi-view stereo approach, implemented in the software Agisoft Photoscan Pro 1.4.3 (Agisoft LCC). The resulting dense point cloud achieved a standard deviation of the location differences between it and the control points of 0.09 m in the horizontal and 0.01 m in the vertical, which is comparable

to other studies using similar ground control point (GCP) densities (e.g. Gindraux et al., 2017). Several studies of factors impacting local photogrammetry-derived Digital Terrain Model (DTM) accuracy (e.g. Gindraux et al., 2017;Tonkin and Midgley, 2016) report a vertical accuracy decrease of ~0.1 m for every 100 m increase in the distance to the closest GCP. In our study the maximum distance from a control point is in the order of 200-300 m. A 6 cm-resolution DTM was produced from the dense point cloud.

The elevation values from a LiDAR dataset collected by the Western Australia Government Department of Biodiversity, Conservation and Attractions in 2012 were subtracted from the UAV DTM produced in this study, to produce a DEM of difference (DoD, Williams, 2012). The average magnitude of the uncertainties associated with the elevation values for the LiDAR dataset is reported as 0.063±0.074 m. The combined location uncertainty in the DoD might therefore be expected to be in the order of 0.1 - 0.2 m. Details of the processing steps are presented in the Supplementary Information.

The focal mechanism computed for the September earthquake (Figure 2) indicates almost pure reverse motion. In this instance, the majority of landscape change would be expected to be vertical, and so detectable in the DoD. In contrast, the November earthquake was dominantly strike-slip (Figure 2). The deformation envelope detectible with the deployed technologies should therefore relate almost exclusively to the September reverse faulting event, with further deformation from the November event remaining largely undetected. To quantify vertical surface displacement associated with the September earthquake, we

extracted swath profiles through the DoD (and the unwrapped InSAR images) parallel and perpendicular to the rupture using swath profiler tool for ESRI ArcGIS (Pérez-Peña et al., 2017). The swath profiles sample the topography perpendicular to the profile to a set distance either side of the profile line. A range of statistics (percentiles, quartiles, max/min/mean) might then be assessed in order to account for the noisy character of the UAV-derived DTM.

**2.3 Rapid deployment aftershock kits**

Within five days of the 16[th] September event five seismic rapid deployment kits (RDKs) and three GPS kits were deployed in the epicentral region. The seismic sensors included four short-period Lennartz LE-3Dlite seismometers and one Trillium Compact seismometer. The short-period instruments were paired with Nanometrics Titan accelerometers recording at 200 Hz





to capture any strong ground motions from the aftershock sequence. The RDK locations were selected to maximise azimuthal coverage of the study area taking into account the location of one permanent Australian National Seismic Network (ANSN)

station near Rocky Gully (RKGY), fortuitously located 24 km from the epicentre of the 16[th] September earthquake.  The network geometry also considered the capture of ground-motion data at a range of source-receiver distances for subsequent attenuation studies.  One RDK (LM01) was deployed at the initial epicentral location of the 16[th] September event.  Waveform data were telemetered in real-time to the National Earthquake Alerts Centre at Geoscience Australia to support real-time aftershock detection and analysis.   The RDK locations therefore required adequate connection to local 3G and 4G

telecommunication networks.  Additional factors considered in the deployment configuration were the local site geology, availability of sunlight to power the sensors, and land access.

The three GPS kits were deployed on existing survey marks within a broader network covering the SWSZ, one of which was last occupied in 2012. The survey marks used (SWSZ45, SWSZ46, SWSZ48) were approximately 36, 65 and 25 km away from the epicentral region, respectively (Figure 2). Processing of the acquired GPS data from the SWSZ46 site produced an

anomalous vertical displacement signal indicating 12 mm of subsidence. This signal could not be reconciled with the surface deformation related to the earthquakes, which was constrained to a near-field region smaller than 10 km from the surface rupture, and is not discussed any further here.

## 3 Results

### 3.1 Field observations

Initial reports from local residents following the September earthquake indicated the presence of west-facing fault scarps intersecting several farm tracks (e.g., Figure 4a), loss of tension in an east-west running fence line (GDA94/MGA50, 479590 mE, 6192140 mN), and cracking in farm dam walls related to lateral spreading (Figure 4b).  Field investigation demonstrated that the track intersections could be linked to form an approximately 3 km-long, concave-to-the-east, west-facing crescentic scarp (Figure 3).  In detail, the scarp comprises a series of left and right stepping *en echelon* segments 100 – 200 m long. In

the central 2 km of the scarp, each segment is associated with up to 20 – 40 cm of vertical displacement (Figure 4c, 4d). The morphology of individual scarp segments show little evidence for a strike-slip component to motion, varying between discrete thrust surface rupture with local fault-parallel folding and cracking, to discrete thrust surface rupture with mole tracks (cf. Lin et al., 2004) and extensional fissuring associated with topographic bulges. Where the scarp crossed drainage lines, presumably associated with metres of sandy alluvial sediments, it becomes a broad warp, often associated with extensional fissuring parallel

to the warp-crest, and warp-perpendicular cracking at step-overs.

A hand trench 2.3 m long and 1.2 m deep was excavated perpendicular to the scarp trace at a location where the vertical displacement was in the order of 0.3 – 0.4 m (Figures 4c & 5). Here, the scarp has an average trend of 025°, and scarp segments are right stepping (GDA94/MGA50, 479290 mE, 6191500 mN). The exposed stratigraphy consists of an approximately 0.10-0.12 m thick organic-rich grey brown silty sand overlying ~0.4 m of pale grey medium sand that becomes pisolitic with depth





(Figure 5b,c). This layer in turn overlies an orange/red mottled grey clayey sand to the bottom of the trench. We interpret the profile to reflect an *in-situ* weathering profile developed in Albany-Fraser Orogen bedrock. Approximately 0.18-0.20 m of the pre-event organic-rich sandy soil had been over thrust along a 20° east-dipping reverse fault that steepened to 30° towards the base of the trench (Figure 5c). A vertical scarp-parallel tension fissure ~0.7 m east of the trench suggests that the fault dip steepens again at shallow depth (Figure 4c). Drag of layering in the hanging wall along the fault resulted in the formation of a

prominent hanging-wall fold. The total vertical deformation at this site is shared approximately equally between discrete slip across the fault plane and folding.

South of 479130 mE, 6191120 mN (GDA94/MGA50) the scarp steps 50 m to the west, the general scarp strike is 355°, and scarp segments become left-stepping. The scarp is associated with a 0.2 – 0.3 m vertical landscape step which extends some 400 m south from this location, before entering dense pine plantation and becoming difficult to follow due to extensive

landscape disturbance (e.g., furrowing), and a thick layer of organic matter cloaking the ground surface (Figure 4d). Within the pine forest, the scarp maintains a vertical displacement of 0.2 – 0.4 m, before stepping again 50 m to the west at 479190 mE, 6190050 mN (GDA94/MGA50).

The segments south of the step-over strike ~340 – 350° and decrease in vertical displacement from ~0.3 m to ~0.1 m at the most southerly intersection of the scarp with a farm track (GDA94/MGA50, 479320mE, 6189440mN). At this track

intersection, observed after the November event, the scarp comprises dominant southeast-trending elements which are locally offset (left-stepping) across east-trending lineaments. These become tensional fissures on the eastern up-thrown side of the scarp. The vertical displacement across the scarp at this location is approximately 0.1 m. Landowners report that there was a 'freshening' [rejuvenation] of the scarp at this location following the November event. The presence of fine cracking details in November, given there were significant rainfall events in October, support these anecdotes. The observations are consistent

with a reverse oblique (sinistral) reactivation of this segment of the scarp. South of the track intersection, the scarp is lost in bushland. However, heavy tree limb fall, and the occasional toppled tree, was observed along strike for a further ~ 400 m to the southeast, suggesting strong localised ground shaking (Figure 3).

### 3.2 Wrapped and unwrapped Sentinel-1 interferograms for both events

The wrapped interferogram for the September event shows an extent of surface deformation ~ 12 km in an east-west direction

and ~ 8 km in a north-south direction (Figure 6a). A linear surface deformation front relating to the rupture can be traced for approximately 5 km. The central ~3 km corresponds to the fault scarp mapped on the ground. The unwrapped interferogram (Figure 6b) shows a broad shallow lobe of negative line of sight (LOS) change that extends from the west to the surface scarp (equivalent to ~2.5 fringes in the wrapped image). From the east, a broad shallow lobe of positive LOS change (~2 fringes in the wrapped image) transitions to a tight (~1.5 km wide) band of negative LOS change characterised by at least 10 fringes.

The images lose coherence in the 200 – 300 m east of the scarp, and in proximity to Lake Noobijup (cf. Figure 3). Coherence is also partly lost beneath an approximately 2 km wide (N-S) easterly trending band of pine forest (see Figures 3 and 6a for location).





The unwrapped interferogram for the September event (Figure 6b) shows a maximum LOS displacement towards the satellite of approximately 0.20 – 0.24 m along the eastern side of the central ~1.5 – 2.0 km of the scarp. These values are approximately
half the maximum scarp height recorded by UAV field measurement, and approximately one third the maximum magnitude of slip along the fault plane that might be calculated from the trench exposure (see also next section). The southern 1.3 km of scarp mapped on the ground occurs mainly beneath pine forest and the unwrapping algorithm failed to reproduce measured scarp heights of 20 – 40 cm in this area. Similarly, the unwrapping algorithm performed poorly in the swampy vegetated land proximal to Lake Noobijup.

The hanging wall uplift diminishes to the east of the scarp and is fringed by a broad region of positive LOS change, corresponding to depression of the land surface. The deepest region of depression of the ground surface is centred on Lake Noobijup (~ 13 cm LOS increase). This area also corresponds to the highest density of aftershocks following the September mainshock (Figure 6b). The depressed region has well-defined northern and southern extents, terminating at east to south-east trending structures. The orientation of these structures suggests that they might have accommodated a significant strike-slip
component to motion, in addition to vertical motion. The areal extent of the depressed region is surprisingly large for a reverse fault rupture (cf. Ellis and Densmore, 2006;King et al., 1988).

The InSAR images for the November event (Figures 6c & d) exhibit the classic quadrupole pattern of an almost pure strike-slip rupture, and are consistent with a maximum of ~5 – 8 cm of right lateral slip having occurred at the surface relating to rupture of a northeast-trending, steeply northwest dipping fault. This contrasts with the focal mechanism for the event, which
suggests an oblique compressive rake (Figure 2). Although the deformation pattern shows a sharp linear discontinuity for ~ 1 km either side of the intersection of the November failure plane with the September rupture plane, simple forward modelling using a finite rectangular elastic dislocation model (Okada, 1985) suggests that a discrete surface rupture may not have been produced (i.e., the rupture tip remained buried). Indeed, structures unambiguously relating to the main rupture were not observed in the field. Potential evidence for secondary surface deformation relating to the main failure plane of this event was
seen as a series of 'fresh' looking east-trending left-stepping dilatational cracks expressed in a boundary track at 478970 mE, 6190390 mN (GDA94/MGA50). No vertical displacement was observed to be associated with these features.

The deformation pattern relating to the November event is also seen to be influenced by the northwest trending structure that forms the southern termination of the September event. This lineament is discernible in the November event unwrapped InSAR from the intersection of rupture planes to approximately 1.8 km southeast (Figure 6d). Minor reverse movement on this
structure during the November event may have been responsible for the reports of a local 'freshening' of the September scarp.

### 3.3 Comparison of UAV-derived and InSAR-derived deformation surfaces

The vertical displacement envelope resulting from the September earthquake was recovered by differencing the UAV DTM and the 2012 LiDAR, producing a DTM of difference (DoD) (Figure 7; see also Supplementary Information). The 1st and 3rd quartile values (Figure 7c) show that within the 100 m-wide swath of the profile, elevation values from the UAV-derived DTM
are consistently within 0.1 m of the LiDAR dataset. In general, the UAV DoD (Figure 7a,c) shows a broad uplifted region to





the east of the scarp trace, which reduces in amplitude relatively smoothly towards the east. Within the ~100 m west of the scarp the land-surface has been depressed into a ~ 0.1 m deep foot-wall depression, before rising again further west. There is a ~0.3 m vertical difference between eastern and western ends of the UAV DoD profile.

Comparison against the InSAR-derived LOS displacement data (cf. Figure 7c blue line and Figures 6a and b) suggests that the eastern end of the DoD profile is located in a broad region of topographic depression (~2 fringes), while the western end is raised relative to the far field (~2.5 fringes). Scarp-parallel and perpendicular profiles through the InSAR-derived LOS displacement data ubiquitously show a smaller displacement magnitude than the vertical deformation estimates from the UAV DoDs (Figure 7b,c).

Along the ~2 km length of scarp that was covered by the UAV DoD, the average height of the scarp, as measured along a 100 m-wide swath profile (Figure 7b), is 0.46 ± 0.11 m.  Note that at the intersection of the scarp-perpendicular profile and the scarp-parallel profile, the scarp-parallel profile under-estimates the true vertical deformation by up to 0.1 m. This is the result of deformation being locally distributed up to 100 – 150 m from the scarp front (e.g., Figure 7c). While the full envelope of deformation has been captured within the uncertainty bounds of the swath measurements, it can be anticipated that spot height

measurements, and to a lesser degree UAV 2-D profiles, might also locally underestimate vertical deformation as variability along and across strike is not sampled (Figure 7b).

No surface expression or ground cracking was observed on the ground along the southeast-trending feature imaged in the InSAR-derived LOS displacement data at the southern end of the September rupture (cf. Figure 6b). However, a DoD constructed over the area (Figure 7a, and Supplementary Information) revealed the presence of a very subtle (≤5 cm high)

linear topographic feature, extending over ~560 m, which was coincident with the surface displacement implied by the InSAR data (Figure 7b).

**3.4 Aftershock relocation**

In the period from 16th September to 23rd November, 884 earthquakes were recorded on stations from the permanent ANSN network and the five rapid deployment aftershock kits temporarily installed in the epicentral region.  Initial locations for these

events were computed using the SeisComP3 seismological software (Weber et al., 2007) and the LocSAT location algorithm (Bratt and Bache, 1988).  *P*- and *S*-wave arrival times were manually picked and reviewed by seismic analysts. To better constrain the location and pattern of aftershocks, events from this dataset were input for relocation using the HypoDD double-difference relative location algorithm (Waldhauser and Ellsworth, 2000), implemented in the software HypoDDpy (Krischer, 2015) (see Supplementary Information for detail). The HypoDD algorithm minimizes errors in hypocentral locations that are

commonly attributed to uncertainties in Earth structure along the event-station ray path. The final relocation included a subset of 470 events from this catalogue.

Epicentres occurring in the interval between the September and November main shocks are located predominantly east of the line of the surface rupture (Figure 6a, b). The associated hypocentres occur in a band between ~1.5 km and 4 km depth, and





below a plane extending from the surface rupture dipping at 30° to the east (Figure 8). Although considerations of the

preservation of seismic moment suggest that the rupture is unlikely to have extended beyond ~ 1.5 km depth, this distribution

is consistent with a slightly steeper dip for the rupture than was indicated by the surface trenching, as suggested by the vertical

tension fissure east of the scarp at the trench location (Figure 4c). At the northern end of the rupture a cluster of hypocentres

is associated with the line-of-intersection of the down-dip extension of the east-dipping rupture plane, and a sub-vertical plane

corresponding at the surface to the east-west trending fault bounding the northern end of the reverse rupture. This is a surprising

finding in that the northern bounding fault is parallel to major east-trending and presumably shallowly south-dipping structures

relating to the northward transport of thrust sheets onto the Yilgarn Craton during the Albany Fraser Orogen (cf. Figure 2). A

linear cluster of hypocentres directly below the southern end of the rupture, parallel to the northwest-southeast oriented

southern terminal structure, suggest that this structure is also steeply dipping. Steep dips for terminal structures at the northern

and southern ends of the reverse rupture is, however, consistent with their orientation sub-parallel to the maximum horizontal

compression direction ($S_{Hmax}$) of the extant crustal stress field (cf. Rajabi et al., 2017).

The central ~5 km of the September rupture, where the most significant surface displacement was observed, was modelled for

Coulomb stress changes (e.g. Lin and Stein, 2004;Toda et al., 2005) (Figure 8, see Supplementary Information for detail of

method and parametrisation). Increases in Coloumb stress are modelled in the footwall at depth, down dip of the rupture plane,

and beyond the tips of the rupture to the north and south. Aftershock hypocentres plot predominantly in a volume down-dip

from the rupture, where the Coulomb stress increased. The concentration of hypocentres at the northern end of the rupture

occur along the line of intersection of the down-dip extension of the east-dipping rupture, and the east-west trending northern

terminal structure, also within a volume of increased Coloumb stress. The greatest area of landscape depression imaged in the

InSAR (Figure 6b), corresponding with Lake Noobijup, occurs above a region of stress relief beyond the trailing edge of the

rupture. If Coloumb stress changes resulting from the September rupture are resolved onto the plane of the November rupture

(i.e., the dextral strike-slip receiver fault) (Figure S3), the plane of the November rupture is seen to be positively stressed over

more than half of its area. The stress increase is approximately 0.4-0.5 MPa in the region of nucleation of the November

earthquake.

The location of the hypocentre of the November $M_W$ 5.2 main shock is consistent with the earthquake nucleating along the line

of intersection between a sub-vertical northeast-southwest trending plane, and the September event rupture plane. This might

potentially explain the poor double couple focal mechanism calculated for the event (Figure 2). Aftershock hypocentres occur

between ~3 km and 4.3 km depth in a volume bound by the northeast-southwest trending, steeply dipping, rupture plane of the

November $M_W$ 5.2 event, and the 30° dipping extension of the September event rupture plane.

**3.5 Relationship between moment magnitude and surface rupture length amongst Australian cratonic earthquakes**

Clark *et al.* (2014b) provided scaling relationships between surface rupture-length and moment magnitude for reverse-faulting

earthquakes occurring in Australian non-extended cratonic settings. Their relationships demonstrate that in this tectonic

setting, earthquakes tend to produce longer earthquake ruptures than may be expected when compared to existing rupture-





scaling relationships (e.g. Wells and Coppersmith, 1994;Leonard, 2014). It is hypothesised that in these regions shallow crustal detachments (Dentith et al., 2000;Drummond et al., 2000) or, alternatively, large shallowly dipping thrust faults (Goleby et al., 1989;Camacho et al., 1995;Korsch et al., 1998) combined with high near-surface stresses (Denham et al., 1980;Denham et al.,

1987) may favour the occurrence of earthquakes at shallow depths. This suggests that relatively narrow fault widths are available for seismogenic rupture, likely leading to large rupture length-to-width aspect ratios for small-magnitude earthquakes. Since the Clark *et al.* (2014b) relationship was presented, the Australian continent has experienced two further surface-rupturing earthquakes with the occurrence of the 2016 Petermann Ranges and 2018 Lake Muir events (see Table 1). Including these new data, the Clark *et al.* (2014b) scaling relationships are reviewed and updated.

With the availability of high-resolution InSAR observations, the extent of earthquake surface deformation can be more readily identified and mapped. Differences between the total "visible" surface rupture lengths (*VSRL*) from field-mapping and "detectable" surface rupture lengths (*DSRL*) from InSAR observations are evident, with the latter observations being longer where these observations are available (Table 1). Consequently, regressions are undertaken between: 1) *VSRL* and $M_W$, and; 2) *DSRL* and $M_W$. The regression takes the following form:

$$SRL = a + b \times M_W \tag{1}$$

where *SRL* is the generic term given to either *VSRL* or *DSRL* and *a* and *b* are coefficients to be determined through regression (Table 3). When *DSRL* from InSAR observations is unavailable, the *DSRL* value is assumed to be equivalent to *VSRL*. It is recognised that this may yield an underestimate of the total rupture length, *DSRL*.

Figure 9a, b shows the least squares relationships relative to other rupture-length scaling relationships used for non-extended

cratonic regions. In general, the updated regressions suggest longer surface ruptures for a given earthquake magnitude, with likely convergence of *DSRL* with Clark *et al.* (2014b) near $M_W$ 7.0.

Because of the likely biases imposed from assuming *DSRL* = *VSRL* for those ruptures that do not have available InSAR observations, a relationship between the ratio of *VSRL-DSRL* and *VSRL* is constructed using limited data from three earthquakes where independent *VSRL* and *DSRL* estimates exist (Figure 9c). It is assumed that earthquakes with decreasing

magnitudes and increasing depth will be expressed at the Earth's surface with decreasing *VSRL* discoverability. The ratio of *VSRL* and *DSRL* could thus be used as a correction factor that could be applied to *VSRL* observations where independent estimates of *DSRL* do not exist. A literature search did not yield any additional data that could be added to the available dataset, particularly for reverse-mechanism ruptures in non-extended cratonic regions. The proposed *VSRL* to *DSRL* correction factor, $\gamma$, can be calculated following:

$$\gamma = \min(\delta, 1.0) \tag{2a}$$

where:

$$\log_{10} \delta = 0.362 \times \log_{10} VSRL - 0.540 \tag{2b}$$

The corrected detectable surface rupture length (*DSRL'*), can thus be determined following:

$$DSRL' = VSRL / \gamma \tag{2c}$$



Supplementing the values of *DSRL*' for events where *DSRL* is not directly measured, revised rupture-scaling relationships between *DSRL*' and $M_W$ can be developed (Figure 9d; Table 3). These revised scaling relationships demonstrate longer rupture lengths for smaller-magnitude earthquakes with lower standard deviation of the residuals (Table 3)

The authors recognise that these relationships are highly conjectural and are based on very limited data. Consequently, the authors invite additional researchers to augment these data to fully scrutinise the legitimacy of the relationships. Nevertheless,

these fault-scaling approaches may have future utility in the improving the characterisation of neotectonic fault scarps and their potential characteristic magnitudes. Should these approaches be refined, they may lead to a decrease in characteristic magnitudes on neotectonic fault scarps in non-extended cratonic regions, such as central and western Australia.

## 4 Discussion

### 4.1 Characteristics of the Lake Muir surface rupture sequence

The comparison of field observations, InSAR imagery, and aftershock earthquake catalogue has permitted exploration of the surface and sub-surface deformation field related to the Lake Muir earthquake sequence in unprecedented resolution. The September rupture, as mapped using traditional paleoseismological means, is revealed to be part of a more extensive deformed region, involving both uplift and depression of broad areas proximal to the surface rupture. Furthermore, the spatial and temporal relationship between September and November $M_W > 5.0$ events reveals a dependency with important implications

for how other earthquake swarms could be interpreted.

### 4.1.1 Reconciling UAV vertical displacement and InSAR LOS displacement measurements

InSAR measures displacement in the one-dimensional LOS of the SAR sensor. Three dimensional displacements of the ground surface are therefore mapped into a one-dimensional geometry. If InSAR data from different viewing geometries (e.g., ascending and descending orbital passes of the SAR satellite) is available, then vertical and horizontal components of

displacement can be resolved (Fuhrmann and Garthwaite, 2019). Unfortunately, this is not the case for the Lake Muir earthquake, where only descending-pass Sentinel-1 SAR data is available. Displacement measurements derived from InSAR analysis of this data must therefore be interpreted in the LOS, and horizontal and vertical signals cannot be unambiguously separated.

This is problematic when attempting to reconcile the single-geometry InSAR LOS data with the absolute elevation changes

captured by the UAV (e.g., Figures 7b and c). However, the descending orbit of the Sentinel-1 data used here has a ground azimuth of 196°, and the SAR sensor looks perpendicularly to the right of this orbit direction (i.e., 286°). The LOS of the SAR sensor is very close to the P-axis of the focal mechanism of the September earthquake (288°). If the earthquake resulted in almost pure thrust motion along the line of the P-axis, as suggested by the focal mechanism (±10°), the single InSAR viewing geometry should be sensitive to the full surface deformation field. The geometry problem is thus reduced approximately to a

vertical plane containing the LOS vector and the slip vector.





We use the above understanding to empirically derive a multiplicative factor of 2.75, which produces a generally good match between the forms of both scarp-perpendicular and scarp-parallel InSAR LOS and UAV vertical displacement profiles (Figure 7b and 7c). It is useful to apply this factor to 'correct' the InSAR profile to enable a comparison of the UAV and InSAR data. There are two locations on the profiles where the match is poor. Firstly, the corrected InSAR LOS profile underestimates the

vertical displacement compared to the UAV profile from 2750 - 3250 m along the profile in Figure 7b. This region corresponds to a 50 m left step in the scarp, and a 25° change in strike direction (Figure 7a). The assumption that the InSAR LOS direction is parallel to the slip vector breaks down here. Secondly, the InSAR profile has not resolved the narrow footwall depression that is apparent in the UAV DoD (Figure 7c). Much of the region occupied by the proximal footwall was masked as the result of lack of coherence of the InSAR signal.

**4.1.2 Character of surface deformation: subsidence in the hanging wall relating to the September event**

The average magnitude of vertical surface deformation along the discrete surface rupture trace, as indicated by the UAV DoD data, and locally validated by the corrected InSAR LOS displacement profile (Figure 7b), is 0.46±0.11 m over the central 2 km of the rupture. The corrected InSAR LOS displacement profile suggests that displacement tails off smoothly to the north from this central plateau (Figure 7b). To the south of the central plateau, there is a step to higher vertical displacement where

the scarp changes orientation (2750 - 3000 m along the profile), and then a fall to the southern extremity. The exact shape of the southern tail, defined by point measurements and 2-D UAV profiles, is largely obscured beneath pine forest. A scarp-perpendicular profile through the UAV and corrected InSAR data (Figure 7c) shows that very little subsidence has occurred in the footwall of the fault (i.e., it is absent in the corrected InSAR data, and seen as a very narrow trough in the UAV DoD), and that the hanging wall uplift relating to the rupture tails off over the ~ 1.0 – 1.5 km to the east of the scarp, falling to a broad

area of subsidence up to ~ 0.3 m below the foot wall level. Consistent with the fault dip measured in the trench (Figure 5c), and deduced from the InSAR correction factor (Figure 7b), elastic dislocation modelling indicates that for fault dips of less than ~35–40°, subsidence should occur above the buried trailing edge of the fault rupture (King et al., 1988;Ellis and Densmore, 2006). A rupture width of <2 km is implied, as compared to the ~5 km subsurface rupture length (not including the southeast segment, Figure 7a).

The rupture is bound to the north and south by steeply dipping highly orthogonal structures that are likely to have accommodated dominantly tear or strike-slip displacement during the September event. These structures bracket the broad region of ground subsidence that occurred to the east of the uplifted hanging wall region (e.g., Figure 6b); an area ~1.5 times that of the uplifted region. The magnitude of the LOS displacement/subsidence is greatest beneath Lake Noobijup (Figure 6b). As field observations failed to find any obvious surface structural development or cracking in this region, the presence of a

subsidence trough may not have been discovered if not for the InSAR data.

In the generally low-relief landscapes typical of intraplate regions, such depressions may have significant impacts on surface and subsurface hydrology. A potential analogue is the 'back-scarp zone' mapped in the hanging wall of the 1968 Meckering surface rupture (Gordon and Lewis, 1980) (see Figure 1; Table 1). The back-scarp zone is an arcuate convex-to-the-east band





of normal faulting and slumping ~ 3 km wide which joined the tips of the concave-to-the-east reverse fault rupture. A single

levelling line across the back-scarp zone identified a 0.3 m depression of the land surface, contrasting to the ~ 1.5 – 2.0 m of throw across the scarp ~ 10 km to the west (Gordon and Lewis, 1980). Changes in hydraulic gradient raised the flood level at Meckering town site by an estimated 12 cm, forcing the relocation of the town to higher ground. In the case of the September Lake Muir event, the hydrology of the important wetland habitat of Lake Noobijup (e.g. Smith, 2010) may be permanently affected by a combination of subsidence, and re-plumbing of the local fractured rock hydrology. There has been no systematic

search to determine if hanging wall subsidence is a characteristic of reverse faults in Phanerozoic eastern Australia (cf. Clark et al., 2012). However, the position of the Wakool and Gunbower fans on the distal hanging wall of the Cadell Fault Block (e.g. Clark et al., 2015) is worthy of investigation in this respect (see Figure 1, location e).

### 4.1.3 Co-location of thrust and strike-slip events

In statistical seismology the uncertainties attached to the calculated locations of small to moderate sized events forming part

of a sequence typically precludes analysis of the detailed temporal, spatial and/or structural relationships between failure surfaces, even with dense instrument networks and sophisticated techniques such as double difference or joint hypocentre relocation (Waldhauser and Ellsworth, 2000). For the Lake Muir sequence the combination of InSAR, field observations and regional aeromagnetic data (cf. Milligan and Nakamura, 2015) provides an unprecedented opportunity to examine the relationships between the two $M_W > 5.0$ events, and thereby gain a better understanding of the distribution of associated smaller

magnitude seismicity.

Intersections between pre-existing basement structures may well have played an important role in nucleating the larger events. The September $M_W$ 5.3 event appears to have nucleated at the intersection between a sub-vertical (or steeply south-dipping) east-trending fault and the ~30° east-dipping main rupture plane. The rupture propagated upward, rupturing the ground surface to produce the observed scarp, and towards the south, ultimately terminating against a sub-vertical northwest-trending

structure. Aftershock hypocentres are located ubiquitously deeper than the trailing edge of the rupture, and below the down-dip extension of the rupture plane. The greatest density of aftershocks, including the largest aftershock ($M_L$ 4.7), occurs at the northern end of the rupture, proximal to the northern bounding fault. This aftershock concentration is spatially coincident with the region of greatest land surface subsidence (Figure 6b). In general, the volume in which aftershocks are located corresponds to a volume of positive Coulomb stress change resulting from the main shock (Figure 8).

As best as can be determined given the uncertainty in the location estimate (i.e., several kilometres), the November $M_W$ 5.2 earthquake nucleated near the centre of the volume of positive Coulomb stress change relating to the September event, potentially along the line of intersection between the down-dip extension of the September rupture and the November rupture plane. A comparison of InSAR images (Figures 6a & c) indicates a spatial coincidence of surface deformation envelopes, with the northeast-striking rupture plane relating to the November event crossing the September rupture near the centre of both

ruptures. The quadrupole deformation pattern for the November event suggests an almost pure strike-slip mechanism, as compared to the oblique compressive focal mechanism for the event (Figure 2, Table 2). A stress condition favourable for





strike-slip failure may have arisen as a result of local stress rotation following the September event. The phase interferogram for the November event also shows a linear fringe boundary coinciding with the southern part of the September rupture (Figure 6c), consistent with the anecdotal evidence for minor reactivation of the rupture. In terms of a simple conceptual block model,

the southern side of the November dextral strike-slip rupture moved to the southwest, and in doing so may have caused a partial reverse reactivation of the September rupture plane. Simultaneous failure along both trends, perhaps originating as a stress concentration at the intersection of the trends (e.g. Talwani, 1988), may explain the 31% double couple for the November event moment tensor. Aftershocks following the November event were confined largely to the volume beneath the down dip extension of the September rupture plain, and southeast of the November rupture plain.

Evidence for the spatial coincidence of similar magnitude moderate sized earthquakes with different failure mechanisms is rare in Australia, typically as the result of large uncertainties in epicentral location estimates. However, the September 2000 to June 2002 Burakin earthquake swarm, which involved approximately 18,000 closely co-located earthquakes (Leonard, 2002) is an exception, and may be considered to be broadly analogous to the Lake Muir sequence. Focal mechanisms were generated for the largest six events of the Burakin swarm, spanning a magnitude range from $M_W$ 4.1 – 4.6 (Leonard et al.,

2002;Allen et al., 2006). These indicate a mixture of thrust and strike-slip ruptures, and one pure normal faulting rupture, all within a ~5 km radius. Preliminary joint hypocentre relocation suggests that these events, and several thousand smaller events, may have originated from as few as 3-4 source areas (Mark Leonard pers. comm., 2018). Aeromagnetic data (Milligan and Nakamura, 2015) indicate that the Burakin swarm area is characterized by a high lineament density, with major trends to the east, northeast and north. Block motion following moderate events may have stressed intersections between these trends,

triggering further events with an eclectic mix of mechanisms.

In central Australia, the reverse mechanism of the 2012 $M_W$ 5.4 Ernabella earthquake (Clark et al., 2014b) was followed fifteen months later by the strike-slip 2013 $M_W$ 5.4 Mulga Park earthquake, which caused extensive surface cracking but no observed surface rupture (Clark and McPherson, 2013). The surface expressions relating to these events indicate that they were likely proximal (≤10 km separation) rather than co-located (Clark and McPherson, 2013). The first event occurred in the hanging

wall of the crustal-scale Woodroffe Thrust Fault (cf. Camacho et al., 1995;Lin et al., 2005;Camacho and McDougall, 2000), and the second event in the foot wall, potentially at a depth of ~10 km (Kevin McCue pers. comm., 2013). Similar to the November Lake Muir event (Figure 2, Table 2), the moment tensor for the Mulga Park strike-slip event had a low percentage double couple, indicating rupture complexity. While it is plausible that the 2012 Ernabella event triggered the 2013 Mulga Park event, it is also possible that in this case the components of the stress field conducive to thrust and strike-slip failure

proximal to the highly structured Woodroffe Thrust crustal 'weakness' (Camacho and McDougall, 2000), were resolved onto spatially discrete structures.

### 4.1.4 Mechanisms for strain localisation in Stable Continental Region (SCR) crust

The September Lake Muir earthquake sequence occurred within rocks and on structures related to the Biranup Zone of the Albany Fraser Orogen, immediately south of the deformed southern margin of the Yilgarn Craton (the 'Northern Foreland'




Spaggiari et al., 2009;Fitzsimons and Buchan, 2005), and sits astride a significant Bouguer gravity anomaly gradient parallel to the boundary (~400 mgal over ~30 km). Major east-trending structures potentially relating to the northward movement of thrust sheets onto the Yilgarn Craton in the Proterozoic form the northern and southern boundaries of the rupture (Figure 2). These structures are cut by a generation of northwest (and subordinate northeast) trending structures, likely steeply dipping and perhaps relating to polyphase tectonism focused on the Darling Fault (cf. Harris, 1996;Olierook et al., 2015)(Figure 1

inset). Elements of these faults are evident in the pattern of ruptures for both Lake Muir $M_W > 5$ events. An obvious question is whether there is anything intrinsic to this crustal and/or structural setting that might predispose the region to moderate to large earthquakes?

At a local scale, the intersection of oblique sets of structures has been proposed as a stress concentrator, potentially leading to seismogenic failure in a critically stressed upper crust (Dentith and Featherstone, 2003;Dentith et al., 2009;Talwani, 1988).

The same geometric complexity of structural elements that may promote initial failure might not be expected to be conducive to continued strain localisation, as the intersections will tend to displace and thus provide a barrier to further slip. This is consistent with the absence of landscape evidence for Late Pleistocene or Holocene rupture on or proximal to the faults that ruptured in the 2018 Lake Muir Sequence (cf. Figure 3). Further research is required to determine if Lake Noobijup has a tectonic origin, given the coincidence of the lake with the region of greatest subsidence during the September earthquake.

At a larger scale, long-term broad-scale regional uplift along the northern margin of the Albany Fraser Orogen is suggested by the youthful relief of the Stirling Ranges (see Figure 1 and Figure 2 inset for location), and evidence for tens of kilometre-scale warping of ferruginous duricrust horizons along the Albany Highway across the axis of the Stirling Ranges (John Myers, GSWA, pers. com. 2002; http://www.ga.gov.au/neotectonic-feature-distribution/home?featureId=442959). A series of lakes along the southern side of the Stirling Ranges, exhibiting progressively increasing elevation and decreasing incision into their

banks towards the west, are also anomalous in this cratonic setting.  While these anomalous landforms have not been studied in detail, from the perspective of causative mechanism or seismogenic potential the east-west axis of uplift (i.e., $S_{Hmax}$ parallel) suggests an alternative to the lithospheric-scale, continental compressive stress field-induced, seismogenic buckling mode of deformation evidenced by the Flinders Ranges (Celerier et al., 2005;Sandiford and Quigley, 2009;Cloetingh et al., 2010;Quigley et al., 2010). A mechanism analogous to the contemporary plate motion driven uplift in the southern Australian

Newer Volcanic field (Joyce, 1975;Davies and Rawlinson, 2014) may be appropriate. In this setting edge-driven mantle convection has resulted in broad, subdued uplift associated with the volcanic field (the Gambier uplift), and of the non-volcanic Padthaway uplift further west, since the Pliocene (e.g. Wallace et al., 2005;Quigley et al., 2010). Similar to the Stirling Ranges, the Padthaway uplift is almost parallel to the regional compression direction, and is not proven to be associated with any topography-forming recurrent faulting (Quigley et al., 2010).

Modelling suggests that uplift and seismogenesis may be a consequence of the structure of the margin itself, and not reliant on plate motion. The MacDonnell and Petermann Ranges in the Northern Territory (Figure 1) are associated with some of the largest Bouguer gravity anomalies in continental settings world-wide, outside regions of active tectonism (Kennett and Iaffaldano, 2013). Detailed interpretation of the gravity anomalies (Aitken et al., 2009 ;Aitken et al., 2009), together with the





analysis of seismic reflection profiles (Korsch and Kositcin, 2010), indicates that the anomalies relate to wedge-shaped zones
bounded by inclined faults, across which the Moho is displaced by up to 15 km (Goleby et al., 1989;Lambeck and Burgess, 1992;Korsch et al., 1998;Hand and Sandiford, 1999). Finite element modelling of the gravitational stresses relating to the crustal architecture across one such fault system, the Redbank Thrust Zone (located along the northern margin of the MacDonnell Ranges – Figure 1), indicates that the region is in broad-scale mechanical equilibrium, despite the local isostatic imbalance (Beekman et al., 1997). However, the models predict the possibility of intraplate seismicity in response to small
changes in tectonic stress, since parts of the crust are already at or close to the failure limit.  In the case of the Redbank Thrust Zone, the application of compressive stress in the form of the modern crustal stress field (Dickinson et al., 2002;Hillis et al., 2008;Rajabi et al., 2017) appears to have perturbed the mechanical equilibrium such that Late Pliocene silcretes have been domed across the axis of the MacDonnell Ranges (Senior et al., 1995), similar to that seen for the duricrust horizons in the Stirling Ranges. Critically, the most highly stressed parts of the Beekman *et al.* (1997) model is the shallow crust in the hanging
wall, and the shallow to mid-crust in the footwall, of the structures controlling the isostatic imbalance. Circumstantially, the 2012 Ernabella earthquake (shallow hanging wall of the Woodroffe Thrust) and 2013 Mulga Park earthquake (moderate depth in the footwall of the Woodroffe Thrust) fit this model (Clark and McPherson, 2013;Clark et al., 2014b), as do the 1986 Marryat Creek (Machette et al., 1993;Crone et al., 1997) and 2016 Petermann Ranges (King et al., 2018;Polcari et al., 2018) ruptures, which occurred in the proximal hanging wall of the Woodroffe Thrust. Similar to the MacDonnell Ranges overlying
the Redbank Thrust Zone, the Petermann Ranges overlying the Woodroffe Thrust are associated with tens of kilometre-scale landscape warping, as evidenced by the deranged Lake Mackay paleo-drainage (Sandiford et al., 2009).

Enhanced seismogenic potential is also modelled to result from variation in the thermal regime in association with significant lithospheric steps (Sandiford and Egholm, 2008), which are ubiquitously associated with the larger gravity anomalies (cf. Kennett et al., 2013;Kennett and Iaffaldano, 2013). The greater seismogenic potential manifests on the side of the boundary
with the thicker lithosphere, by virtue of thermal weakening. The 2001-2002 Burakin earthquake swarm could potentially be attributable to this mechanism (Leonard, 2002;Allen et al., 2006). However, counter examples are provided by the paleo-earthquake scarps and earthquake swarms evenly distributed across the Yilgarn Craton (Clark, 2010;Dent, 2016). So, while a crustal/lithospheric architecture argument *may* be invoked to explain the Lake Muir earthquake sequence, such an argument is not universally applicable across the Precambrian SCR crust of Australia (cf. Figure 1).

**4.2 One-off ruptures from moderate to large magnitude earthquakes in the cratonic regions of Australia**

Conceptually, for the purposes of probabilistic seismic hazard assessment, a *fault source* is a seismogenic fault that has produced earthquakes in the past, and can be expected to continue doing so (Musson, 2012). Excluding the November $M_W$ 5.2 event, for which a unique surface rupture could not be mapped in the field, the September $M_W$ 5.3 Lake Muir earthquake was the ninth event documented to have produced surface rupture in Australia in historical times (Figure 1, Table 1). These ruptures
are located exclusively in the Precambrian SCR crust of central and western Australia (Figure 1), and none could have been identified and mapped using topographic signature prior to the historical event (Table 1). For example, Crone et al. (1997)



excavated trenches across the 1986 Marryat Creek and 1988 Tennant Creek ruptures and found that while each rupture in part exploited pre-existing bedrock faults, there was no unequivocal geomorphic, stratigraphic or structural evidence to suggest a penultimate event in the preceding 50-100 kyr or more. A similar conclusion was made on the basis of trenching investigations

of the 1968 Meckering surface rupture (see Clark and Edwards, 2018, and references therein).

Paleoseismological investigations of several faults in the same Precambrian SCR tectonic setting provide evidence for limited recurrence of large earthquakes, with up to four events documented on an individual fault within the last *ca.* 100 kyr (Crone et al., 2003;Clark et al., 2008;Estrada, 2009). These scarps - Roopena, Hyden, Lort River and Dumbleyung (see Figure 1 for locations) - all overlie simple through-going faults imaged in aeromagnetic data (Milligan and Nakamura, 2015). The two-to-

five Quaternary events documented on the Hyden (Clark et al., 2008) and Lort River (Estrada, 2009) scarps are all that are evident across Late Neogene duricrust. While shallow trenches across the 2-5 m high Roopena scarp exposed Precambrian bedrock on both sides of the fault (Crone et al., 2003), nearby scarps are associated with an extended Neogene to Recent history of movement (McCormack, 2006;Miles, 1952;Weatherman, 2006). For example, the Randell and Poynton Faults on the northeastern Eyre Peninsula (South Australia) are associated with 30-70 m of Pliocene and younger vertical displacement

(McCormack, 2006). Scarps developed in the *ca.* 15 Ma surface of the Nullarbor Plain (Figure 1), which overlies Neoproterozoic mobile belt basement, are associated with up to 15-30 m of vertical surface displacement (Clark et al., 2012;Hillis et al., 2008), implying the recurrence a dozen or so neotectonic events per fault at most.

In general, scarps developed within Archean and Paleoproterozoic crust tend to be more modest in height, less well connected (i.e., spatially isolated), and more complex in plan than scarps in Mesoproterozoic and Neoproterozoic crust (Clark et al.,

2012). The remarkable sequence of three 'one-off' surface breaking earthquakes in 1968, 1970 and 1979 (Meckering, Calingiri and Cadoux — Gordon and Lewis, 1980;Lewis et al., 1981) raises interesting questions about the potential for multiple modes of upper crustal failure when stress thresholds are exceeded. The structurally 'complex' Meckering, Calingiri and Cadoux scarps (Figure 1) are 70–100 km apart; too distant for static stress changes to have promoted rupture (cf. Caskey and Wesnousky, 1997). Furthermore, the ruptures were sufficiently temporally separated that dynamic stress changes are unlikely

to have promoted rupture (e.g. Belardinelli et al., 2003). The observations are consistent with the postulate that blocks of upper crust on the scale of ~$10^4$ square kilometres can unload in the space of a decade (Clark et al., 2012). In the case of southwest Western Australia, this process may be facilitated by a mid- to upper-crustal architecture characterised by fundamental sub-horizontal structural discontinuities (most notably at ~10 km and ~25 km depth) (Dentith et al., 2000;Drummond and Mohamed, 1986;Everingham, 1965;Goleby et al., 1993) that are compartmentalised by major moderately-dipping fault

systems, forming "super terranes" as envisaged by Wilde et al. (1996). Perhaps the presence of through-going faults perpendicular to the crustal stress field may mean the difference between an unloading scenario involving strain localization in successive events on a single fault (e.g., Dumbleyung), or on several proximal structures (e.g., Meckering). As mentioned in the previous section, intersection of structural trends, and rheological changes at the boundaries of mafic dykes have been proposed as stress concentrators, potentially promoting 'weakness' (Dentith and Featherstone, 2003;Everingham and Gregson,

1996). A pattern is emerging where 'one-off' ruptures, as evidenced by the historic surface-breaking earthquakes (Figure 1,





Table 1) are filling the spaces between mapped multi-event neotectonic scarps. With few examples, it is not possible to draw conclusions with any certainty.

This preponderance of 'one-off' events suggests caution in applying a traditional elastic strain accumulation model to Precambrian SCR crust (cf. Clark, 2010;Braun et al., 2009). Indeed, over the last few decades, permanent and campaign GPS
studies have failed to detect a tectonic deformation signal from which a strain budget could be calculated across all of Australia (e.g. Tregonning, 2003). Similar studies have used these observations, amongst others (e.g. Calais et al., 2005), to propose that one-off events and clusters of large events either deplete long-lived pools of 'fossil' lithospheric stress (Liu and Stein, 2016;Calais et al., 2016) and/or that there is an orders of magnitude difference in the timescales of elastic strain accumulation and seismogenic strain release (e.g. Clark et al., 2015;Craig et al., 2016). By virtue of the scarcity of data with which to
validate such a model, the underpinning assumption that a 'long-term slip rate' is a meaningful concept in an intraplate setting, as per the prevailing plate margin paradigm, remains to be fully tested. Indications are that the concept may be useful in the Phanerozoic stable continental region (SCR) crust of eastern Australia (Figure 1), where faults with up to a few hundreds of metres of neotectonic slip occur (cf. Clark et al., 2015;Clark et al., 2017). However, it is not so certain whether assigning a long-term slip rate is meaningful for Precambrian SCR crust (e.g. Calais et al., 2016). In the absence of meaningful recurrence
of large events, building relations for fault displacement hazard using rupture traces from cratonic Australia is fraught (cf. Boncio et al., 2018).

As was the case with the 2016 Petermann Ranges earthquake rupture (King et al., 2018;Polcari et al., 2018;Wang et al., 2019), the Lake Muir September event surface rupture was longer than might have been expected from scaling relationships between magnitude and surface rupture length (e.g. Wells and Coppersmith, 1994;Clark et al., 2014b;Leonard, 2014). New relationships
developed as part of this study (Figure 9) allow for the distinction between "visible" surface rupture lengths (*VSRL*) from field-mapping, and "detectable" surface rupture lengths (*DSRL*) from increasingly more readily available InSAR data that more closely define the complete rupture extent. Relative to the scaling relationships introduced by Clark et al. (Clark et al., 2014b; equivalent to the VSRL relationships presented here), the updated relationships include more data and yield lower uncertainties, particularly when the *DSRL* is used. In general, the *DSRL* scaling relationships will yield longer ruptures for a
given event magnitude, converging with the Clark et al. (2014b) scaling relationships near $M_W$ 7.0. Users should exercise caution extending these relationships to lower magnitudes where the intersection of the rupture plane with the surface becomes less likely. Nevertheless, the relatively large rupture lengths observed for moderate-to-large earthquakes in non-extended Australian cratonic crust challenges notions that SCR earthquakes should yield higher stress drops (e.g., Allmann and Shearer, 2009), with smaller rupture areas (e.g., Brune, 1970). Conservation of high stress drop for SCR events, commensurate with
Allmann and Shearer (2009), would require narrow down-dip rupture widths that yield large aspect ratios. The surface displacement field revealed in the InSAR suggests that the rupture width is < 2.0 km, suggesting non-uniform scaling between rupture length and width. Systematic analysis of stress drop for recent moderate-to-large ($M_W \geq 5.0$) Australian earthquakes should be undertaken to test the nature of stress drop relative to surface rupture to provide further constraint on the expected rupture dimensions of SCR earthquakes in Australia.



### 4.3 Migration of the locus of moment release in the Southwest Seismic Zone

Pre-historical 'one-off' ruptures cannot generally be detected using widely available and areally extensive digital elevation data (e.g. SRTM, Clark, 2010). Irrespective, neotectonic fault displacements are typically < 5 m across much of the Precambrian SCR (Clark and Allen, 2018). An upper limit on the long-term rates at which the displacement has accrued is provided by regional bedrock erosion rates of < 5 m/Myr (Stone et al., 1994;Belton et al., 2004;Jakica et al., 2010). Displacement accumulation rates over the last 100 kyr can be as high as 30 m/Myr (Clark et al., 2008). Such rates are clearly unsustainable over the long term given the non-mountainous character of the landscape, leading researchers to propose that displacement is highly episodic in this setting (e.g. Crone et al., 1997;Crone et al., 2003;Clark et al., 2008). This finding is consistent with an analysis of the expected relief generation rates by Leonard & Clark (2011) which imply that the historical catalogue of seismicity in the Southwest Seismic Zone (SWSZ) is ten times that required to build the scarps (see also Braun et al., 2009). However, revised earthquake rate estimates based on the remediation of catalogue magnitudes for the NSHA18 (Allen et al., 2018) suggest that long-term forecasts of large-earthquake rates in Precambrian crust may have been previously overestimated, which would provide an improved correspondence between the historical and pre-historical earthquake records. Nevertheless, the question remains as to whether the seismic activity within a given area containing neotectonic fault scarps is episodic on individual faults and so migratory within a region (e.g. Liu and Stein, 2016), or whether the locus of seismicity migrates over time, never to return to previously active regions/faults, as suggested by the paleoseismology of the nine historical surface ruptures.

A plot of earthquake epicentres since 1960 colour coded by age (Figure 10), shows the central core of events that were used by Doyle (1971) to define the extents of the SWSZ. Much of this activity occurred in the two-to-three decades including and following the Meckering, Calingiri and Cadoux surface-rupturing earthquakes. These epicentres are enclosed by a broad ring of more recent epicentres, framing the core on northern, eastern and southern sides. The epicentres, including rare ~$M_W$ 5 events (e.g., Lake Muir, Burakin), typically fill the interstices between the paleoseismic fault scarps (cf. Clark, 2010). Rarely can epicentres be confidently associated spatially with paleoseismic fault scarps (e.g. within 10 km). The pattern might be interpreted as a cascading destabilisation radiating outwards from the 1968-1979 surface ruptures. While having produced surface rupture in one instance (e.g., Lake Muir), it is not clear whether the destabilisation might be expected to trigger future $M_W > 6$ events, or if the larger earthquakes occur in response to an as yet unknown alternative mechanism.

Including eight of the nine earthquakes mentioned above, forty-two onshore earthquakes of magnitude $M_W \geq 5.0$ are located in the Australian Precambrian cratonic crust (cf. Allen et al., 2018). Given that hypocentres are typically extremely shallow in the cratonic areas (Clark et al., 2014a;Leonard, 2008), these events may also have ruptured the ground surface, in which case a landscape record may exist and could be investigated to help understand their structural context.





### 4.4 Future use of InSAR for earthquake studies in Australia

There has been slow uptake of InSAR for earthquake studies in Australia. This is due in part to incomplete SAR data archives available for analysis, and the absence of earthquakes large and/or shallow enough to be detectable by InSAR. Dawson *et al.* (2008) were able to use L-band ALOS-PALSAR interferometry to study the deformation field of two very small earthquakes, but due to poor SAR data availability before and after earthquakes, and interferometric baseline restrictions, it was not possible to study all earthquakes that had occurred.

The C-band Sentinel-1 mission has been a game-changer for InSAR applications because of its near-global coverage and consistent repeat-imaging strategy. The availability of this rich data set has enabled global systematic studies of earthquake detectability in InSAR data for the first time (e.g. Barnhart et al., 2019;Funning and Garcia, 2019). Following the launch of Sentinel-1A and Sentinel-1B in 2014 and 2016 respectively, there are now regular SAR acquisitions over the whole Australian continent every 12 days (from descending-pass orbits). This means that co-seismic image pairs of 12 days are consistently available, which ensures that interferograms have a good chance of maintaining a coherent phase signal. This will result in many more Australian earthquakes being detected with InSAR in the future.

A limitation of the C-band InSAR is that it does not perform well in all land-cover environments. Funning & Garcia (2019) showed that the main control on whether an earthquake is detectable by InSAR is governed by vegetation density and long spatial baselines. Although the latter is well-controlled with Sentinel-1 (with spatial baselines not usually exceeding a couple hundred metres), vegetation density is variable across the Australian continent. Magnitude and depth also play a role in earthquake detectability with InSAR (Dawson and Tregoning, 2007;Funning and Garcia, 2019). Large magnitude earthquakes ($Mw \geq$ 6-7) produce surface deformation patterns over larger areas that are easier to detect, although localised decorrelation can occur near the rupture zone due to intense ground movement (e.g. Wang et al., 2019). Smaller magnitudes ($M_W \leq 6$) may be difficult to accurately detect due to the smaller spatial extent of deformation patterns. Furthermore, earthquakes that are shallower (< 5 km) are more likely to have a surface expression, while deeper ones may not.

### 5 Conclusions

A shallow $M_W$ 5.3 earthquake near Lake Muir in southwest Western Australia on the 16th of September 2018 was followed on the 8th of November by a co-located $M_W$ 5.2 event. Focal mechanisms produced for the events suggest reverse and strike-slip rupture, respectively. Recent improvements in the coverage and frequency of SAR data over Australia with the Sentinel-1 satellite constellation has allowed for the timely mapping of the surface deformation fields relating to both earthquakes in unprecedented detail. Field mapping, guided by the InSAR data, reveal that the first event produced an approximately 3 km-long and up to 0.4-0.6 m high west-facing surface rupture, consistent with slip on a moderately east-dipping fault. Interpretation of InSAR data shows that the surface scarp relates to a sub-surface rupture ~5 km long, bound at its north and southern extremities by strike-slip terminal structures. New data, and the recognition that InSAR data will increasingly allow for the distinction between "visible" surface rupture lengths (*VSRL*) from field-mapping and "detectable" surface rupture lengths



(*DSRL*), has prompted a recalculation of the Clark *et al.* (2014b) relation between rupture length and magnitude for SCR earthquakes. The regressions indicate that Australian SCR earthquakes tend to be longer for a given magnitude than elsewhere in the world (e.g. Wells and Coppersmith, 1994;Leonard, 2014).

The September $M_W$ 5.3 Lake Muir earthquake was the ninth event documented to have produced surface rupture in Australia in historical times (Figure 1, Table 1). These ruptures are located exclusively in the Precambrian SCR rocks of central and western Australia, and none could have been identified and mapped using topographic signature prior to the historical event. A pattern is also emerging where 'one-off' ruptures, as evidenced by the historic surface-breaking earthquakes, are filling the spaces between mapped multi-event neotectonic scarps (cf. Clark, 2010;Clark et al., 2012). Despite such observations, patterns

of migration in intraplate seismicity, particularly in Precambrian SCR crust, are yet to be fully understood (Liu et al., 2011;Stein and Liu, 2009).

**Data availability**

Source parameters of the earthquakes were obtained from the Geoscience Australia catalog https://earthquakes.ga.gov.au/ (last accessed 08.02.2019). The original and relocated datasets are obtainable from the Geoscience Australia Github repository

https://github.com/GeoscienceAustralia/GA-neotectonics/tree/master/Lake_Muir_Solid_Earth_data. Focal mechanisms for the three largest events were obtained from https://earthquake.usgs.gov/earthquakes/ as per Table 2. The datafile and code written to regress the length versus magnitude data, and the UAV DTM of difference GeoTIFF files are also obtainable from the Geoscience Australia Github repository, using the above link. The DCBA LiDAR dataset was used under licence and cannot be provided to third parties. The precise orbital ephemerides products used in correcting the InSAR data are available

from https://qc.sentinel1.eo.esa.int/aux_poeorb/.

**Author contribution**

DC and SS designed the field experiments and executed them. GB relocated the seismicity data and wrote related sections of the manuscript. TA developed the regression code for assessing relationships between moment magnitude and surface rupture

length and wrote related sections of the manuscript. SB acquired and processed the InSAR data. MG assisted SB in writing up related sections of the manuscript. DC prepared the manuscript with contributions from all co-authors.

**Competing interests**

The authors declare that they have no conflicts of interest.



**Acknowledgements**

Thanks to Roger Hearn of Manjimup for conducting the initial field reconnaissance and taking the first photos of the scarp. Thanks also to Rob De Campo and Mark Muir for providing access to their properties to conduct fieldwork. Guorong Hu of Geoscience Australia kindly processed the GPS data. Jasmine Rutherford from the Western Australia DBCA is warmly thanked for providing access to their LiDAR data. This manuscript is published with the permission of the CEO of Geoscience

Australia.

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





**Tables**

| Event | Year | Magnitude (Mw) | Mapped surface rupture length (km)# | Maximum vertical displacement (m) | Reference |
|-------|------|----------------|-------------------------------------|-----------------------------------|-----------|
| Meckering, WA | 1968 | 6.58 | 37 | 2.5 | Gordon & Lewis (1980); Clark & Edwards (2018) |
| Calingiri, WA | 1970 | 5.46 | 3.3 | 0.4 | Gordon & Lewis (1980) |
| Cadoux, WA | 1979 | 6.13 | 14 | 1.4 | Lewis *et al.* (1981) |
| Marryat Creek, SA | 1986 | 5.74 | 13 | 0.9 | Machette *et al.* (1993) |
| Tennant Creek, NT* | 1988 | 6.76 | 36 | 1.8 | Crone e*t al.* (1992, 1997) |
| Katanning, WA | 2007 | 4.73 | 0.2 (1.26) | 0.1 | Dawson *et al.* (2008) |
| Ernabella, SA | 2012 | 5.37 | 1.5 | 0.5 | Clark *et al.* (2014b) |
| Petermann Ranges, NT | 2016 | 6.10 | 15 (20) | 1.0 | King *et al.* (2018); Polcari et al. (2018); Wang et al. (2019) |
| Lake Muir, WA | 2018 | 5.30 | 3.2 (7) | 0.4 | this article |

**Table 1: Historical earthquake events known to have produced surface rupture in Australia (expanded after Clark et al., 2014b). * The Tennant Creek surface rupture was produced by three events in a 12 hr period (Bowman, 1992). # Values in brackets reflect the surface rupture length estimated from InSAR data.**






| Date | Latitude | Longitude | Depth (km) | Depth uncertainty (km) | $M_W$ | $M_L$ | Event page (USGS = moment tensors) |
|---|---|---|---|---|---|---|---|
| 16/09/2018 | -34.3897 | 116.7988 | 0.02 | 1.92 | 5.32 | 5.71 | https://earthquakes.ga.gov.au/event/ga2018sfzeme  https://earthquake.usgs.gov/earthquakes/eventpage/us2000hfcw |
| 13/10/2018 | -34.39523 | 116.79883 | 5.77 | 1.90 | | 4.65 | https://earthquakes.ga.gov.au/event/ga2018ucpciy |
| 9/11/2018 | -34.42316 | 116.78733 | 3.02 | 1.46 | 5.19 | 5.40 | https://earthquakes.ga.gov.au/event/ga2018wamvnf  https://earthquake.usgs.gov/earthquakes/eventpage/us1000hpej |

**Table 2: Selected source parameters and recourse links for the three largest events in the Lake Muir earthquake sequence.**

| Regression* | a | a (std err) | b | b (std err) | Std |
|---|---|---|---|---|---|
| *VSRL* (LSQ) | -5.38 | 0.788 | 1.064 | 0.135 | 0.250 |
| *VSRL* (ODR) | -5.76 | 0.815 | 1.130 | 0.140 | 0.169 |
| *DSRL* (LSQ) | -3.58 | 0.733 | 0.777 | 0.126 | 0.233 |
| *DSRL* (ODR) | -3.91 | 0.753 | 0.833 | 0.129 | 0.181 |
| *DSRL'* (LSQ) | -2.97 | 0.480 | 0.693 | 0.082 | 0.152 |
| *DSRL'* (ODR) | -3.10 | 0.485 | 0.716 | 0.083 | 0.125 |

**Table 3. Coefficients between *SRL* (*VSRL* or *DSRL* in km) and $M_W$ for substitution into Equation 1. * the regression type used is either least squares (LSQ) or orthogonal distance (ODR) regression, respectively.**





**Figure Captions**

**Figure 1: Neotectonic features (red lines) from the Australian Neotectonic Features Database (Clark et al., 2012;Clark, 2012). Historical surface ruptures shown as red dots labelled with the year of the event. Base map shows neotectonic superdomains (after**
**Leonard et al., 2014), and the outlines of Australian States and Territories. Note all historical surface ruptures have occurred in Precambrian stable continental region (SCR, Johnston et al., 1994) crust.**

**Figure 2: Location and geological setting of the 2018 Lake Muir earthquake sequence. Location inset shows basement geology modified after the GSWA 1:500 000 State interpreted bedrock geology of Western Australia, 2016 (Geological Survey of Western Australia, 2016). Proterozoic and younger faults and dykes are shown as thick and thin black lines respectively. Base map in the**
**main frame shows reduced to pole total magnetic intensity data (Milligan and Nakamura, 2015). Black triangles show the locations of seismic recording stations, and GPS stations (SWSZ prefix). The location of the September earthquake is shown by a white star, with its associated uncertainty ellipse (see Table 2 for source). Surface ruptures from the September and November events are shown by black solid and dashes lines, respectively (see text for details). Focal mechanisms are shown for the two M>5 events (see Table 2 for source).**

**Figure 3: Map of the Lake Muir surface ruptures and associated seismicity. Background map is part of the Western Australian Department of Biodiversity, Conservation and Attractions LiDaR holdings (https://www.dbca.wa.gov.au/contact-us) with ESRI world imagery overlain at 75% transparency. White line shows the extent of the discrete surface rupture relating to the September event, as mapped from InSAR. Superposed black lines are scarp segments mapped on the ground. Grey dashed line shows the discrete line of deformation relating to the November event, as mapped from InSAR. The three largest events, 16th September $M_W$5.3,**
**13th October $M_L$4.6, and 9th November $M_W$5.2, are consecutively labelled.**

**Figure 4: Photographs of the surface rupture: (a) 13 cm high scarp crossing farm track. Photo taken by Roger Hearn on 27/09/2018. Looking north (479101 mE, 6190727 mN); (b) east-trending tension fissures relating to lateral spread into a farm dam (479588 mE, 6192126 mN); (c) 40 cm high scarp and hanging wall tension fissure at the hand trench location (see Figure 5). Looking north (479285 ME, 6191496 mN); (d) 40 cm high scarp in pine plantation. Looking northeast (479112 mE, 6190422 mN). GD94/MGA50.**

**Figure 5: Hand trench location (479285 ME, 6191496 mN GDA94/MGA50). (a) subset of the photogrammetrically-derived UAV DTM (see Figure 3 for full footprint), with topographic section indicated by the black line. Colour drape has been tilted to remove regional topographic slope and enhance relative differences; (b) photomosaic of the north wall of the hand trench showing folded and displaced strata; and (c) interpretation of stratigraphy and structure of the north wall of the hand trench.**

**Figure 6: Phase images and images of the unwrapped InSAR line of sight (LOS) displacement field for the (a) & (b) September**
**$M_W$5.3 and (c) & (d) November $M_W$5.2 events. The location of the surface rupture relating to the September event is shown as a white line, with a black dashed line showing where the scarp was observed in the field. The surface deformation front relating to the November event is shown as a dashed black line. Refer to Figure 2 for focal mechanisms. Seismicity before and after the November event is shown black dots on parts (a) & (b) and (c) & (d), respectively. The main shocks are shown as red stars. Each fringe in (a) and (c) represents 2.8 cm of LOS range change. Note several unwrapping errors are evident as regions bound by a step jump at the**
**northern and southern end of the scarp in part (b).**





**Figure 7: Relative co-seismic displacement swath profiles through UAV and InSAR data; (a) Location of scarp parallel and perpendicular swath profiles. Mapped scarp elements are shown in red. Length-weighted rose diagrams (north up) show orientation of scarp elements. The extent of the UAV surveys are shown as black outlines. UAV DTM differenced against LiDaR is shown over main scarp segment (blues are small, greys are large). Base map is the InSAR phase image for the September event over LiDaR**
**(datum/projection = GDA94/MGA50); (b) Scarp-parallel profiles. UAV profiles sample a 100 m wide swath centred on the profile line and plot relative vertical displacement. The 90th percentile minus the 10th percentile value elevation value is plotted to reduce noise in the UAV data resulting from the difficulty in removing low vegetation inherent to structure from motion DTMs. UAV 2-D profiles plot the vertical displacement measured from single profile lines through small area UAV DTMs flown along forest trails. Spot heights were measured visually with a tape measure where the forest was too dense for other techniques to be used. InSAR**
**swath profiles are co-located with the UAV profile lines and sample 300 m either side of the profile line. These plot relative LOS displacement, calculated as the difference between the maximum and the minimum value in each scarp-perpendicular swath. (c) 1st to 3rd quartile range of a scarp-perpendicular swath profile through the UAV data is plotted along with the mean value to demonstrate the precision of the UAV data. Variation in LOS displacement for the co-located unwrapped InSAR profile is not resolvable at the scale of the figure, so the maximum value sampled by each scarp-parallel swath is plotted. Zero relative**
**displacement is arbitrarily pinned to the eastern end of the profile in part.**

**Figure 8: Coulomb stress changes resulting from the September M$_W$ 5.3 event (plan and section). The rupture area was modelled as being 5.0 km long by 2.0 km wide to constrain seismic moment and slip, and 30° east-dipping consistent with the USGS focal mechanism for the event (Figure 2, Table 2,). Relocated seismicity is overlain onto the plan and projected onto the vertical A-B section plane (see Figure 3 for seismicity legend). Black line is the surface deformation trace interpreted from ground observations and InSAR data. Black arrows in the upper panel show a linear trend of epicentres that relate to the November M$_W$ 5.3 strike-slip**
**event.**

**Figure 9: The least squares relationships for (a) VSRL and (b) DSRL relative to other magnitude-rupture length scaling relationships, including Wells and Coppersmith (1994; surface rupture length and sub-surface rupture length, respectively), Clark et al (2014b) and Leonard (2014; surface rupture length and fault rupture length for SCR dip-slip events, respectively). (c) The ratio**
**of VSRL and DSRL plotted against VSRL used to determine the postulated correction factor used to calculate DSRL' as shown in Equation 2. (d) Corrected DSRL' values based on adjustment factors in Equation 2.  Note, data points where DSRL = VSRL are plotted for reference, but are not used to determine the magnitude scaling coefficients in Table 3.**

**Figure 10: A plot of Southwest Seismiz Zone (SWSZ) earthquake epicentres since 1960, colour coded by age, showing outward migration from the centres at Meckering, Calingiri and Cadoux. Neotectonic fault scarps (Clark, 2012) are shown as black lines.**
**Seismographs are identified by the following abbreviations. AU = Australian National Seismic Network (Geoscience Australia), IU = Global Seismograph Network (IRIS/USGS), S = Australian Seismometers in Schools (ANU).**





**Figures**




**Figure 1**







**Figure 2**





**Figure 3**





**Figure 4**





**Figure 5**



Figure 6





**Figure 7**




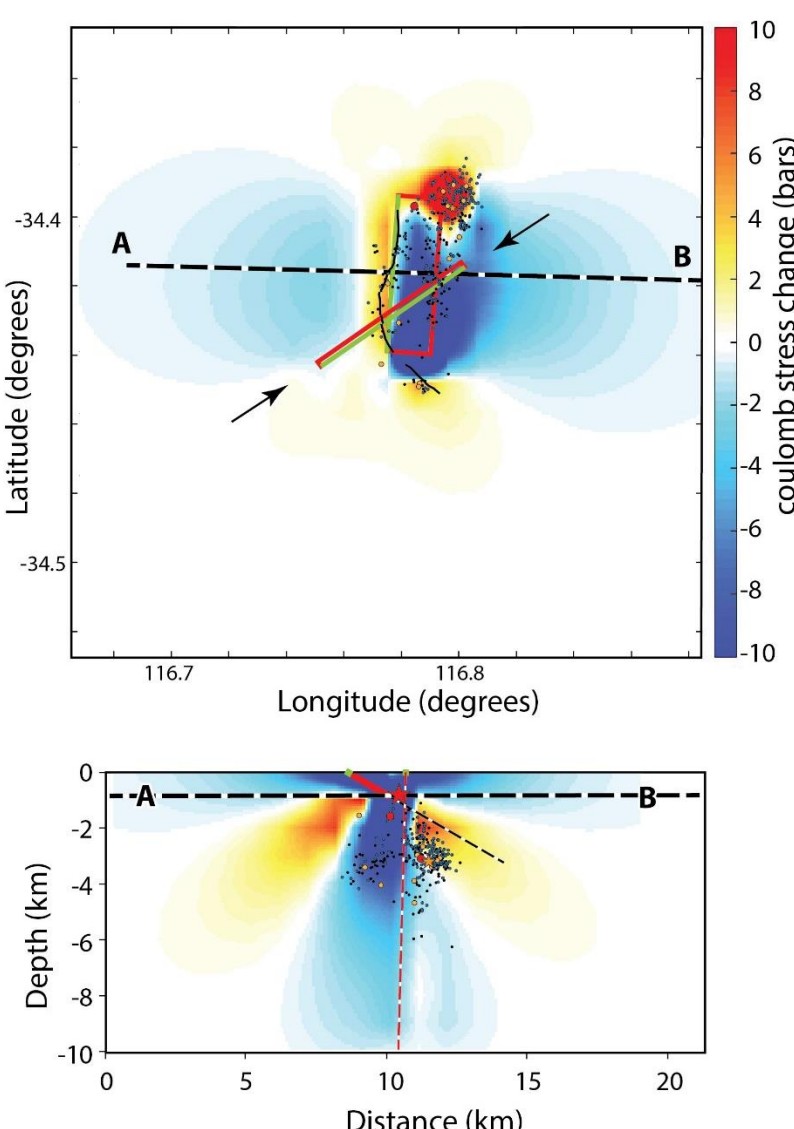

**Figure 8**





**Figure 9**





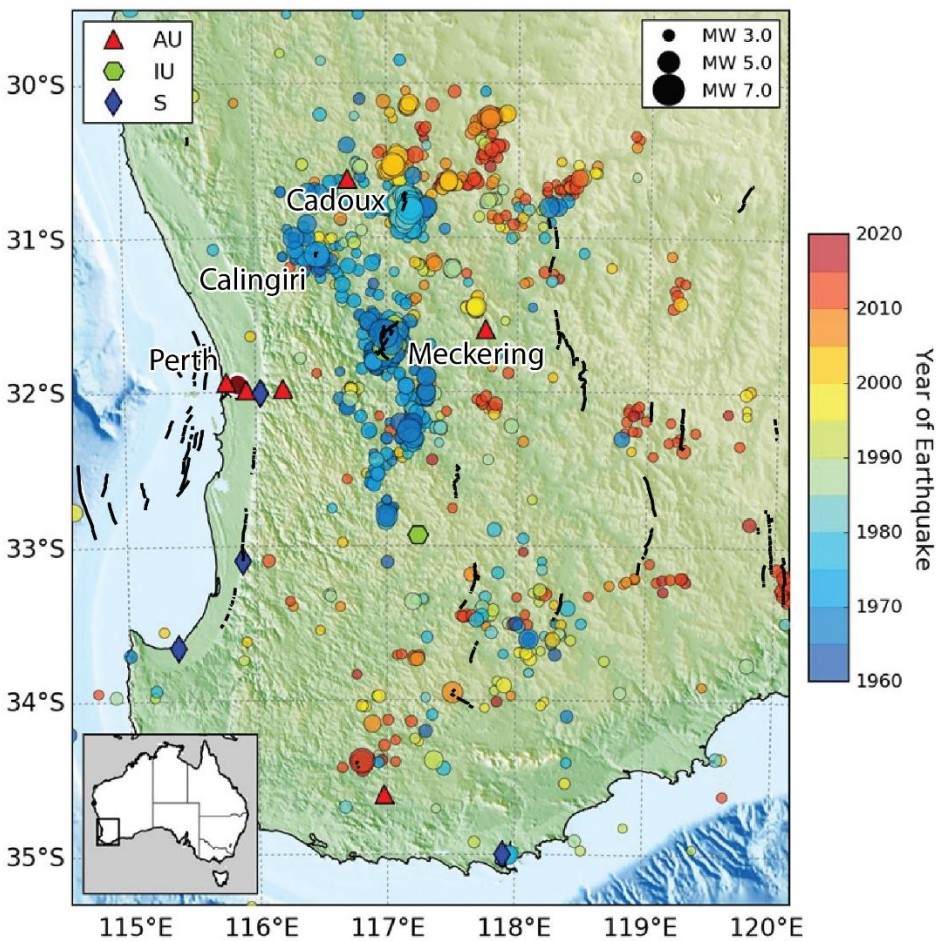

**Figure 10**
