# Peer review of "Surface deformation relating to the 2018 Lake Muir earthquake sequence, southwest Western Australia: new insight into stable continental region earthquakes"

_Solid Earth, 2019_

## Referee Comment (RC1) · Anonymous Referee #1 · 23 Oct 2019

Dear Editor, the purpose of the submitted paper by Clark et al. "The 2018 Lake Muir earthquake sequence, southwest Western Australia: rethinking Australian stable continental region earthquakes" matches the aim of the SOLID EARTH magazine.

General Overview I think the Authors made a very good work analyzing the two earthquake sequences using, at the same time, satellite SAR interferograms, seismic locations and field mapping. In addition, very interesting appear the comparison among the behavior of the Lake Muir sequence and the previous others sequences, and the analysis of the other earthquakes that produce surface rupture in SW Australia. These facts make this paper enough innovative for its multidisciplinary approach to the analysis of

the earthquakes location areas and their related structures.

Conversely, the paper carries a lot of different discussions and results, passing from the multidisciplinary approach (SAR Interferometry, field observations and seismicity) of the two seismic sequences and their relationship with the surface ruptures and deformation rates, up to the discussion of the need to introduce new relationships between moment magnitude and surface rupture length for the SCR craton area. In my opinion this latter argument deserves a separate paper. The paper should be more focused to the expected results introduced by the title. Following this choice, the paper, after some focused minor revisions, might be acceptable for publication.

In the following points, I carried over any observations about the discussion on the text:

• Paragraph "Rapid deployment aftershock kits" and Figure 2: Regarding the seismic station deployment, the closest station has been located at least at 24 km far from the epicentral area of both earthquakes. Although the magnitude of two main shocks are greater than 5 and their location uncertainties could be low, however large uncertainties might afflict the location of small magnitude earthquakes occurred during the swarms. These uncertainties have implications about the matching with the surface ruptures found by the field activities and with the INSAR interferometry computation. These critical points are responsibly highlighted by the Authors, anyway they represent "low-resolution" results.

• Lines 445-448: The Authors describe the fact the November event is located in the positive Coulomb stress change induced by the September earthquake and show the location of the aftershocks swarms in relation to the faults and surface ruptures. The don't clearly explain if they consider the November earthquake induced by a dynamic triggering of the September event, but they only describe the matching with the INSAR interferograms and the rotation of local stress after the September earthquake, in order to facilitate the strike-slip failure of the November earthquake. I think the clarification of this point could be an important statement for future and more addressed studies, for

instance the fault ruptures interaction or the dynamic triggering between two or more seismic sources.

• In figure 6 the Authors show both the wrapped and unwrapped interferograms. This choice allows the reader to better appreciate the measures observed by an interferogram. However, in Figure 6a there is an east-west oriented fringe interruption at latitude/longitude 6190000/4790000? How the Authors interpret it?? • Finally, the text contains a lot of place names. For a "not Australian" reader is a bit hard to follow the text with the lack of a clear tectonic map containing the bright place names tags.

In particular: Line 508: ÂăFlinders Ranges is missing in the figures. Line 540: Burakin earthquake area is missing in the figures.

—————————————————

---

## Editor Comment (EC1) · Cristiano Collettini (Editor) · 15 Nov 2019

Dear Authors, I have received one review for your manuscript plus 11 answers of colleagues that declined to review it, some of them declined also after having accepted to look through the manuscript. In order to keep the review process relatively short I reviewed the manuscript and in the following I am presenting my comments on the manuscript.

The present manuscript uses an interdisciplinary approach to study the 2018 Lake Muir earthquake sequence occurred in southwest Western Australia. The sequence consists of two Mw 5.3 and 5.2 mainshocks and associated aftershocks. The Authors

analyze the sequence mainly via interferometric wide swath SAR images, field mapping of the scarps produced by the events and earthquakes relocation and analysis. While the characterization of the seismic sequence and associated surface deformation is innovative and very interesting, the manuscript in its entire is quite confusing since the reader cannot properly understand which is the main scientific message the paper seeks to deliver: from one side there is a description of the 2018 Lake Muir earthquake sequence, and from another side there is a re-assessment on Australian seismicity. This point has been also raised by Referee 1. To improve the manuscript and make it a potential contribution for Solid Earth I suggest focusing on the characterization of the seismic sequence, its surface deformation and propose some explanation for the longer ruptures in comparison to the commonly used scaling relationships. In order to improve this part, I also suggest making a comprehensive figure where data are integrated to give the possibility to the reader to pick-up a collective picture of the entire dataset.

In the following I am highlighting some aspects that should be improved.

Title: Change in the title the part dealing with rethinking Australian stable continental region Earthquakes.

Introduction: this is an introduction focused on Australian earthquakes, lake Muir sequence, its geological and morphological settings. For an international journal it would be more appropriate an introduction presenting data on moderate magnitude seismic sequences, their associated surface deformation and surface breaks (length). At line 96 the Authors say: "LiDAR dataset (see Supplementary Information) revealed the presence of grain in the landscape". Is it possible to better explain the meaning of grain in the landscape or add a reference for it?

Location of seismicity. As also pointed out by Referee 1, large uncertainties might affect the location of small magnitude earthquakes occurred during the sequence and therefore some sentences are not properly supported by data. Here you are some examples. At lines 285-295 the Authors infer fault geometry by aftershock distribution

(e.g. Figure 6 and 8). To me this dataset is not enough to depict fault geometry, some good examples of fault geometry from aftershock distribution are presented in (Waldhauser et al., GRL 2004; Valoroso et al., Geology 2014; Shelly et al., JGR 2016; Chiaraluce et al., SRL, 2017).

Paragraph 3.5 Relationship between moment magnitude and surface rupture length amongst Australian cratonic earthquakes. In Figure 9.c there are only 3 data, the other panels build on Clark et al., 2014 BSSA where Length vs. M for Australian earthquakes are plotted in figure 11, and the Authors themselves at the end of the paragraph say: "The authors recognise that these relationships are highly conjectural and are based on very limited data. Consequently, the authors invite additional researchers to augment these data to fully scrutinise the legitimacy of the relationships". I suggest removing this paragraph.

Discussion

Paragraph 4.1.3 Co-location of thrust and strike-slip events: This is a quite big speculation since the resolution of the data do not allow for this, or data are not well presented to convince the reader about this. Provide an integrated picture to support the co-location. The sentence starting at line 443 and saying: "In general, the volume in which aftershocks are located corresponds to a volume of positive Coulomb stress change resulting from the main shock (Figure 8)", is not 100% consistent with aftershock distribution. I suggest to significantly reduce this part and incorporate it in the discussion on the seismic sequence.

Paragraph 4.1.4 Mechanisms for strain localisation in Stable Continental Region (SCR) crust. This paragraph is not strongly related to the data presented in the manuscript but mainly based on literature. I suggest removing this part.

Paragraph 4.2 One-off ruptures from moderate to large magnitude earthquakes in the cratonic regions of Australia. This paragraph mainly builds on earthquakes occurring in 68, 70 79, not extensively discussed in the present manuscript and as the Authors

themselves say at line 586: "With few examples, it is not possible to draw conclusions with any certainty". In addition the final part of the paragraph is highly speculative since I agree with the Authors that systematic analysis of stress drop for recent moderate-to-large (MW $\geq$ 5.0) Australian earthquakes should be undertaken to test the nature of stress drop relative to surface rupture to provide further constraint on the expected rupture dimensions of Precambrian stable continental region earthquakes in Australia. I suggest removing this part.

Paragraph 4.3 Migration of the locus of moment release in the Southwest Seismic Zone. The data presented in the manuscript: a) a seismic sequence with 2 M larger than 5 earthquakes, and b) a distribution of the seismicity from 1960 for SW Australia, are not enough to provide a solid scientific background supporting the migration of the locus of seismic moment release. I suggest deleting it.

4.4 Future use of InSAR for earthquake studies in Australia. This paragraph is more likely a technical report for scientists interested in using InSAR for earthquakes studies in Australia and therefore I consider it inappropriate as the final paragraph of the discussion. I suggest removing this part.

---

## Author Comment (AC1) · 19 Dec 2019

General:

We agree with the reviewers that our contribution has attempted to cover a lot of ground, some not exclusively related to the new data presented on the Lake Muir earthquake sequence. With this in mind we have trimmed the manuscript discussion. However, we retain insight that the Lake Muir sequence has contributed to regarding stable continental region earthquakes. Further, we expand the introduction to appeal more to the international readership of Solid Earth, and better scope our stable continental region focus. One area of concern for both reviewers was the degree to which

deductions on earthquake genesis and relationships to faulting may be made from our main shock locations and aftershock relocations. We recognise that the uncertainties relating to earthquake locations were not well-communicated, and have rectified this deficiency. Specifically, we note that the uncertainties on aftershock locations are better than 300 m in all cases. The reviewed manuscript presented the initial Australian National Seismic Network locations for the three largest events in the sequence as these events failed key tests for double-difference relocation. This resulted in the undesirable situation where the main shocks were associated with horizontal location uncertainties of 5-6 km. In the revised manuscript we have relocated the two largest events based upon the relatively well located third largest event in the sequence. This relocation has resulted in collapse of the horizontal location uncertainty ellipses to $\sim$ 1 km, and allows for better comparison between main shocks, aftershocks, and surface and geological data. The revised Figures 3 and 6 are attached as an example of the improvement.

Reply to specific comments made by RC1:

• Section 2.3. "Rapid Deployment aftershock kits... Regarding the seismic station deployment, the closest station has been located at least at 24 km far from the epicentral area of both earthquakes... large uncertainties might afflict the location of small magnitude earthquakes occurred during the swarms". The reviewer misunderstood our communication of the experimental design. The nearest permanent network station is 24 km from the epicentral area. This is explicitly stated in the text. The five rapid deployment aftershock kits range in distance from right on top of the first main shock (LM01), to 42 km distant (LM05). We have added text to Section 2.3, and the caption of Figure 2, to make clear that the black triangles with labels prefixed by 'LM' on Figure 2 are the temporary stations. We understand that the network, comprising permanent and temporary stations, is not as dense as might be achieved in regions where there is a higher perception of earthquake hazard (i.e. non-SCR), but are satisfied that our uncertainties (better than 300 m for aftershocks) is suitable to make our conclusions. •

Section 3.5. "In my opinion this latter argument [new relationships between moment magnitude and surface rupture length] deserves a separate paper. The paper should be more focused to the expected results introduced by the title." This section has been removed as suggested by both reviewers. • Line 445-448. "They don't clearly explain if they consider the November earthquake induced by a dynamic triggering of the September event ...I think the clarification of this point could be an important statement for future and more addressed studies, for instance the fault ruptures interaction or the dynamic triggering between two or more seismic sources.' Excellent point! The text has been modified to reflect the fact that we do indeed think that the second M5 event was triggered. Further, comparison to the other 'swarm events' in Australia which comprise M5 events suggests that mechanical interaction of 'blocks' results in subsequent proximal (or co-located) triggered events. We demonstrate in this article that the triggered events can have different failure mechanisms. • "in Figure 6a there is an east-west oriented fringe interruption at latitude/longitude 6190000/4790000? How the Authors interpret it?" The text of Section 3.2 states "Coherence is also partly lost beneath an approximately 2 km wide (N-S) easterly trending band of pine forest (see Figures 3 and 6a for location)". The north-south extent of this forest is clearly marked with an arrow labelled "pine forest" on both Figures 3 and 6a. We have added the following text to the caption of Figure 6: The north-south extent of an easterly trending band of pine forest associated with degradation of coherence is indicated with a white arrow in part (a). • "For a "not Australian" reader is a bit hard to follow the text with the lack of a clear tectonic map containing the bright place names tags." Figure 1 is a clear tectonic map as far as stable continental region crust may be divided with respect to seismogenic potential (e.g. Johnston et al 1994; Clark et al. 2012). In the revised manuscript, we have either included all Australian place names mentioned in the text in Figure 1, or written explicit locations into the text itself.

**Earthquake epicentres (magnitude ML)**

- 0.33 - 1.00
- 1.01 - 2.00
- 2.01 - 3.00
- 3.01 - 4.00
- 4.00 - 5.70

L. Noobijup

north extent
of observed scarp

trench

south extent
of observed scarp

tree damage

pine forest

N

// observed September EQ scarp

line of rupture of November EQ

UAV mission footprint

linear valley trend

0 1 2 3 4 km

6194000
6192000
6190000
6188000

476000 478000 480000 482000 484000

**Fig. 1.** Revised Figure 3: Map of the Lake Muir surface ruptures and associated seismicity.

**Fig. 2.** Revised Figure 6: Phase images and images of the unwrapped InSAR line of sight (LOS) displacement field for the (a) & (b) September MW5.3 and (c) & (d) November MW5.2 events.

---

## Author Comment (AC2) · 19 Dec 2019

General:

We agree with the reviewers that our contribution has attempted to cover a lot of ground, some not exclusively related to the new data presented on the Lake Muir earthquake sequence. With this in mind we have trimmed the manuscript discussion. However, we retain insight that the Lake Muir sequence has contributed to regarding stable continental region earthquakes. Further, we expand the introduction to appeal more to the international readership of Solid Earth, and better scope our stable continental region focus. One area of concern for both reviewers was the degree to which

deductions on earthquake genesis and relationships to faulting may be made from our main shock locations and aftershock relocations. We recognise that the uncertainties relating to earthquake locations were not well-communicated, and have rectified this deficiency. Specifically, we note that the uncertainties on aftershock locations are better than 300 m in all cases. The reviewed manuscript presented the initial Australian National Seismic Network locations for the three largest events in the sequence as these events failed key tests for double-difference relocation. This resulted in the undesirable situation where the main shocks were associated with horizontal location uncertainties of 5-6 km. In the revised manuscript we have relocated the two largest events based upon the relatively well located third largest event in the sequence. This relocation has resulted in collapse of the horizontal location uncertainty ellipses to $\sim 1$ km, and allows for better comparison between main shocks, aftershocks, and surface and geological data. The revised Figures 3 and 6 are attached as an example of the improvement.

Reply to specific comments made by EC1:

• "I suggest focusing on the characterization of the seismic sequence, its surface deformation and propose some explanation for the longer ruptures in comparison to the commonly used scaling relationships". The focus of the manuscript has been reframed in the introduction, and the relationship developed between visible surface rupture length (VSRL) and detectible surface rupture length (DSRL) removed as suggested. We instead consider the relationship of our new results to existing empirical scaling relationships, and propose possible explanations. • "Comprehensive figure where the data are integrated". It is not clear what the reviewer desires here. Figure 6 presents the InSAR results as a base, with the relocated epicentres, and rupture traces from field mapping and InSAR (black and white lines respectively) overlain. The only data not presented on this figure are the UAV data, which are presented in Figure 7. We feel that combining the UAV data with Figure 6 would unnecessarily clutter the figure. Or does the reviewer refer to his comment on Section 4.1.3, which is addressed

below? • "Title: Change in the title the part dealing with rethinking Australian stable continental region Earthquakes": Title text after the colon modified to more concisely introduce the paper content: "new insight into Australian stable continental region earthquakes" • "For an international journal it would be more appropriate an introduction presenting data on moderate magnitude seismic sequences, their associated surface deformation and surface breaks (length)". We have rewritten the Introduction to better set the scope of the manuscript. We retain our focus on the stable continental region (SCR) setting, but have presented our study in the framework of global SCR earthquakes that precede the Lake Muir sequence, many of which have been imaged by InSAR. Table 1 has been expanded considerably to present data relating to these earthquakes. • "Is it possible to better explain the meaning of grain in the landscape or add a reference for it?". We mean grain in the same way that trees have grain; a preferential alignment of constituent elements. To clarify, the word 'grain' in the sentence has been replaced with "an alignment of valleys and ridges". • "At lines 285-295 the Authors infer fault geometry by aftershock distribution. . . To me this dataset is not enough to depict fault geometry". As a preamble to addressing this reviewer concern we note that in response to reviewer RC1 misunderstanding our aftershock deployment geometry we have added text to section 2.3 and to the caption of Figure 2 noting that the rapid deployment kits are prefixed by LM on Figure 2. We have also stated the uncertainties associated with the aftershock relocations in the text of Section 3.4, which was not made clear before. The mean location uncertainties in the relocated dataset were calculated to be 63 m, 116 m, and 228 m in the east, north and up directions, respectively. It is true that the computed aftershock distribution does not define a neat rupture plane, nor are relocated earthquakes so numerous, as may be the case in the plate margin examples presented by the reviewer. However, given the tight uncertainties we contend that the scatter is real, and the main concentration of aftershocks 'in general' occupy a volume defined by positive coulomb stress changes. As the reviewer notes, this relationship is not 100%. This stems in part from the 'real' scatter, but mainly from the limitations in depicting the
3-D aftershock cloud in 2-D sections. The hypocentres occurring at depth immediately beneath the rupture plane, within a volume of modelled coulomb stress decrease, mostly relate to the November MW 5.2 event. This is, however, not exclusively the case, implying the presence of foreshocks on the November rupture plane. We have tightened the text of Section 3.4, and the caption of Figure 8, to improve the clarity of communication of our observations. • Section 3.5. "Relationship between moment magnitude and surface rupture length amongst Australian cratonic earthquakes. I suggest removing this paragraph". As this section was of concern to both reviewers, we have removed it. Instead, we consider the relationship of our new results to existing empirical scaling relationships, and propose possible explanations, in the discussion. • Section 4.1.3. Co-location of thrust and strike-slip events: This is a quite big speculation since the resolution of the data do not allow for this, or data are not well presented to convince the reader about this. Provide an integrated picture to support the co-location. We must conclude that there are deficiencies in our presentation of the data as the evidence from the INSAR images is compelling and incontrovertible for an overlap of surface deformation envelopes resulting from the two largest events. This is explicitly stated in the text and is shown in Figure 6, where the surface trace of the strike-slip fault rupture relating to the November event (panels c and d) has been superposed onto the surface deformation pattern of the September event (panels a and b). We agree that there is significant uncertainty in relating the main shocks and aftershock distribution to geological structures. We have reworded the section to recognise that this is secondary evidence supporting the primary correlation using InSAR. • Section 4.1.3. "The sentence starting at line 443. . . In general, the volume in which aftershocks are located corresponds to a volume of positive Coulomb stress change resulting from the main shock (Figure 8)", is not 100% consistent with aftershock distribution. . . I suggest to significantly reduce this part and incorporate it in the discussion on the seismic sequence". As mentioned above, we have tightened the text in terms of communicating the uncertainties associated with the hypocentral locations. Further, we have provided explanation in the text for the hypocentres that

do not occur in the volume of positive Coloumb stress (these mostly relate to the November event). We contend that this justifies retaining the section. • Section 4.1.4 "Mechanisms for strain localisation in Stable Continental Region (SCR) crust... This paragraph is not strongly related to the data presented in the manuscript but mainly based on literature. I suggest removing this part." We have trimmed this section to focus more on the immediate region of the Lake Muir sequence, then tie this to material presented on global stable continental earthquake mechanisms in the reworded introduction. To remove this section entirely would be to lose a discussion of the insight that the events might give to the setting of SCR earthquakes globally. • Section 4.2 "One-off ruptures from moderate to large magnitude earthquakes in the cratonic regions of Australia... I suggest removing this part." The section has been shortened and reworded to emphasise that there is no evidence for prior rupture on the Lake Muir faults, and this is typical of Precambrian SCR crust, as presented in the reworded introduction. • Section 4.3. "Migration of the locus of moment release in the Southwest Seismic Zone. . . I suggest deleting it" This section has been deleted as suggested. A few sentences have been incorporated into the revised section 4.2 for clarity of argument. • Section 4.4. Deleted as suggested.

Please also note the supplement to this comment:
https://www.solid-earth-discuss.net/se-2019-125/se-2019-125-AC2-supplement.pdf
* * *
**Earthquake epicentres (magnitude ML)**

- 0.33 - 1.00
- 1.01 - 2.00
- 2.01 - 3.00
- 3.01 - 4.00
- 4.00 - 5.70

L. Noobijup

north extent
of observed scarp

trench

south extent
of observed scarp

tree damage

pine forest

// observed September EQ scarp

line of rupture of November EQ

UAV mission footprint

linear valley trend

N

0  1  2  3  4
km

476000  478000  480000  482000  484000

6194000  6192000  6190000  6188000

**Fig. 1.** Revised Figure 3: Map of the Lake Muir surface ruptures and associated seismicity.

**Fig. 2.** Revised Figure 6: Phase images and images of the unwrapped InSAR line of sight (LOS) displacement field for the (a) & (b) September MW5.3 and (c) & (d) November MW5.2 events.

**Supplement:**

**Author response to reviewers** (our words in red)

General:

We agree with the reviewers that our contribution has attempted to cover a lot of ground, some not exclusively related to the new data presented on the Lake Muir earthquake sequence. With this in mind we have trimmed the manuscript discussion. However, we retain insight that the Lake Muir sequence has contributed to regarding stable continental region earthquakes. Further, we expand the introduction to appeal more to the international readership of Solid Earth, and better scope our stable continental region focus. One area of concern for both reviewers was the degree to which deductions on earthquake genesis and relationships to faulting may be made from our main shock locations and aftershock relocations. We recognise that the uncertainties relating to earthquake locations were not well-communicated, and have rectified this deficiency. Specifically, we note that the uncertainties on aftershock locations are better than 300 m in all cases. The reviewed manuscript presented the initial Australian National Seismic Network locations for the three largest events in the sequence as these events failed key tests for double-difference relocation. This resulted in the undesirable situation where the main shocks were associated with horizontal location uncertainties of 5-6 km. In the revised manuscript we have relocated the two largest events based upon the relatively well located third largest event in the sequence. This relocation has resulted in collapse of the horizontal location uncertainty ellipses to ~ 1 km, and allows for better comparison between main shocks, aftershocks, and surface and geological data. The revised Figures 3 and 6 are attached as an example of the improvement.

Reply to specific comments made by EC1:

- "I suggest focusing on the characterization of the seismic sequence, its surface deformation and propose some explanation for the longer ruptures in comparison to the commonly used scaling relationships". The focus of the manuscript has been reframed in the introduction, and the relationship developed between visible surface rupture length (VSRL) and detectible surface rupture length (DSRL) removed as suggested. We instead consider the relationship of our new results to existing empirical scaling relationships, and propose possible explanations.
- "Comprehensive figure where the data are integrated". It is not clear what the reviewer desires here. Figure 6 presents the InSAR results as a base, with the relocated epicentres, and rupture traces from field mapping and InSAR (black and white lines respectively) overlain. The only data not presented on this figure are the UAV data, which are presented in Figure 7. We feel that combining the UAV data with Figure 6 would unnecessarily clutter the figure. Or does the reviewer refer to his comment on Section 4.1.3, which is addressed below?
- "Title: Change in the title the part dealing with rethinking Australian stable continental region Earthquakes": Title text after the colon modified to more concisely introduce the paper content: "new insight into Australian stable continental region earthquakes"
- "For an international journal it would be more appropriate an introduction presenting data on moderate magnitude seismic sequences, their associated surface deformation and surface breaks (length)". We have rewritten the Introduction to better set the scope of the manuscript. We retain our focus on the stable continental region (SCR) setting, but have presented our study in the framework of global SCR earthquakes that precede the Lake Muir sequence, many of which have been imaged by InSAR. Table 1 has been expanded considerably to present data relating to these earthquakes.
- "Is it possible to better explain the meaning of grain in the landscape or add a reference for it?". We mean grain in the same way that trees have grain; a preferential alignment of constituent elements. To clarify, the word 'grain' in the sentence has been replaced with "an alignment of valleys and ridges".

- "At lines 285-295 the Authors infer fault geometry by aftershock distribution... To me this dataset is not enough to depict fault geometry". As a preamble to addressing this reviewer concern we note that in response to reviewer RC1 misunderstanding our aftershock deployment geometry we have added text to section 2.3 and to the caption of Figure 2 noting that the rapid deployment kits are prefixed by LM on Figure 2. We have also stated the uncertainties associated with the aftershock relocations in the text of Section 3.4, which was not made clear before. The mean location uncertainties in the relocated dataset were calculated to be 63 m, 116 m, and 228 m in the east, north and up directions, respectively. It is true that the computed aftershock distribution does not define a neat rupture plane, nor are relocated earthquakes so numerous, as may be the case in the plate margin examples presented by the reviewer. However, given the tight uncertainties we contend that the scatter is real, and the main concentration of aftershocks 'in general' occupy a volume defined by positive coulomb stress changes. As the reviewer notes, this relationship is not 100%. This stems in part from the 'real' scatter, but mainly from the limitations in depicting the 3-D aftershock cloud in 2-D sections. The hypocentres occurring at depth immediately beneath the rupture plane, within a volume of modelled coulomb stress decrease, mostly relate to the November $M_W$ 5.2 event. This is, however, not exclusively the case, implying the presence of foreshocks on the November rupture plane. We have tightened the text of Section 3.4, and the caption of Figure 8, to improve the clarity of communication of our observations.
- Section 3.5. "Relationship between moment magnitude and surface rupture length amongst Australian cratonic earthquakes. I suggest removing this paragraph". As this section was of concern to both reviewers, we have removed it. Instead, we consider the relationship of our new results to existing empirical scaling relationships, and propose possible explanations, in the discussion.
- Section 4.1.3. Co-location of thrust and strike-slip events: This is a quite big speculation since the resolution of the data do not allow for this, or data are not well presented to convince the reader about this. Provide an integrated picture to support the co-location. We must conclude that there are deficiencies in our presentation of the data as the evidence from the INSAR images is compelling and incontrovertible for an overlap of surface deformation envelopes resulting from the two largest events. This is explicitly stated in the text and is shown in Figure 6, where the surface trace of the strike-slip fault rupture relating to the November event (panels c and d) has been superposed onto the surface deformation pattern of the September event (panels a and b). We agree that there is significant uncertainty in relating the main shocks and aftershock distribution to geological structures. We have reworded the section to recognise that this is secondary evidence supporting the primary correlation using InSAR.
- Section 4.1.3. "The sentence starting at line 443... In general, the volume in which aftershocks are located corresponds to a volume of positive Coulomb stress change resulting from the main shock (Figure 8)", is not 100% consistent with aftershock distribution... I suggest to significantly reduce this part and incorporate it in the discussion on the seismic sequence". As mentioned above, we have tightened the text in terms of communicating the uncertainties associated with the hypocentral locations. Further, we have provided explanation in the text for the hypocentres that do not occur in the volume of positive Coloumb stress (these mostly relate to the November event). We contend that this justifies retaining the section.
- Section 4.1.4 "Mechanisms for strain localisation in Stable Continental Region (SCR) crust... This paragraph is not strongly related to the data presented in the manuscript but mainly based on literature. I suggest removing this part." We have trimmed this section to focus more on the immediate region of the Lake Muir sequence, then tie this to material presented on global stable continental earthquake mechanisms in the reworded introduction. To remove this section entirely would be to lose a discussion of the insight that the events might give to the setting of SCR earthquakes globally.
- Section 4.2 "One-off ruptures from moderate to large magnitude earthquakes in the cratonic regions of Australia... I suggest removing this part." The section has been shortened and

reworded to emphasise that there is no evidence for prior rupture on the Lake Muir faults, and this is typical of Precambrian SCR crust, as presented in the reworded introduction.

- Section 4.3. "Migration of the locus of moment release in the Southwest Seismic Zone… I suggest deleting it" This section has been deleted as suggested. A few sentences have been incorporated into the revised section 4.2 for clarity of argument.
- Section 4.4. Deleted as suggested.

Reply to specific comments made by RC1:

- Section 2.3. "Rapid Deployment aftershock kits… Regarding the seismic station deployment, the closest station has been located at least at 24 km far from the epicentral area of both earthquakes... large uncertainties might afflict the location of small magnitude earthquakes occurred during the swarms". The reviewer misunderstood our communication of the experimental design. The nearest *permanent* network station is 24 km from the epicentral area. This is explicitly stated in the text. The five rapid deployment aftershock kits range in distance from right on top of the first main shock (LM01), to 42 km distant (LM05). We have added text to Section 2.3, and the caption of Figure 2, to make clear that the black triangles with labels prefixed by 'LM' on Figure 2 are the temporary stations. We understand that the network, comprising permanent and temporary stations, is not as dense as might be achieved in regions where there is a higher perception of earthquake hazard (i.e. non-SCR), but are satisfied that our uncertainties (better than 300 m for aftershocks) is suitable to make our conclusions.
- Section 3.5. "In my opinion this latter argument [new relationships between moment magnitude and surface rupture length] deserves a separate paper. The paper should be more focused to the expected results introduced by the title." This section has been removed as suggested by both reviewers.
- Line 445-448. "They don't clearly explain if they consider the November earthquake induced by a dynamic triggering of the September event …I think the clarification of this point could be an important statement for future and more addressed studies, for instance the fault ruptures interaction or the dynamic triggering between two or more seismic sources.' Excellent point! The text has been modified to reflect the fact that we do indeed think that the second M5 event was triggered. Further, comparison to the other 'swarm events' in Australia which comprise M5 events suggests that mechanical interaction of 'blocks' results in subsequent proximal (or co-located) triggered events. We demonstrate in this article that the triggered events can have different failure mechanisms.
- "in Figure 6a there is an east-west oriented fringe interruption at latitude/longitude 6190000/4790000? How the Authors interpret it?" The text of Section 3.2 states "Coherence is also partly lost beneath an approximately 2 km wide (N-S) easterly trending band of pine forest (see Figures 3 and 6a for location)". The north-south extent of this forest is clearly marked with an arrow labelled "pine forest" on both Figures 3 and 6a. We have added the following text to the caption of Figure 6: The north-south extent of an easterly trending band of pine forest associated with degradation of coherence is indicated with a white arrow in part (a).
- "For a "not Australian" reader is a bit hard to follow the text with the lack of a clear tectonic map containing the bright place names tags." Figure 1 *is* a clear tectonic map as far as stable continental region crust may be divided with respect to seismogenic potential (e.g. Johnston et al 1994; Clark et al. 2012). In the revised manuscript, we have either included all Australian place names mentioned in the text in Figure 1, or written explicit locations into the text itself.

---

## Author Response (AR1)

**Author response to reviewers (our words in red)**

Preamble:

Firstly, I would like to sincerely apologise for the delay in returning the revised manuscript. We have had a horror summer in Canberra (eastern Australia), involving drought, bushfire, hailstorms, and flood. This has severely disrupted all our lives (https://www.abc.net.au/news/2020-03-01/canberras-summer-of-fire-heat-rain-and-hail/12011124). I thank you for your patience.

As suggested, I have attached the marked-up version of the revised manuscript to this document. The reviews, while concise, suggested a rather extensive revision. Consequently, there is a lot going on in the marked-up document. For this I apologise also.

General:

We agree with the reviewers that our contribution has attempted to cover a lot of ground, some not exclusively related to the new data presented on the Lake Muir earthquake sequence. With this in mind we have trimmed the manuscript discussion. However, we retain insight that the Lake Muir sequence has contributed to regarding stable continental region earthquakes. Further, we expand the introduction to appeal more to the international readership of Solid Earth, and better scope our stable continental region focus. One area of concern for both reviewers was the degree to which deductions on earthquake genesis and relationships to faulting may be made from our main shock locations and aftershock relocations. We recognise that the uncertainties relating to earthquake locations were not well-communicated, and have rectified this deficiency. Specifically, we note that the uncertainties on aftershock locations are better than 300 m in all cases. The reviewed manuscript presented the initial Australian National Seismic Network locations for the three largest events in the sequence as these events failed key tests for double-difference relocation. This resulted in the undesirable situation where the main shocks were associated with horizontal location uncertainties of 5-6 km. In the revised manuscript we have relocated the two largest events based upon the relatively well located third largest event in the sequence. This relocation has resulted in collapse of the horizontal location uncertainty ellipses to ~ 1 km, and allows for better comparison between main shocks, aftershocks, and surface and geological data. The revised Figures 3 and 6 are attached as an example of the improvement.

Reply to specific comments made by EC1:

- "I suggest focusing on the characterization of the seismic sequence, its surface deformation and propose some explanation for the longer ruptures in comparison to the commonly used scaling relationships". The focus of the manuscript has been reframed in the introduction, and the relationship developed between visible surface rupture length (VSRL) and detectible surface rupture length (DSRL) removed as suggested. Discussion of the length of ruptures has been reserved for a future publication. Instead we consider in more detail the treiggering implications between events, as suggested by reviewer RC1.
- "Comprehensive figure where the data are integrated". It is not clear what the reviewer desires here. Figure 6 presents the InSAR results as a base, with the relocated epicentres, and rupture traces from field mapping and InSAR (black and white lines respectively) overlain. The only data not presented on this figure are the UAV data, which are presented in Figure 7. We feel that combining the UAV data with Figure 6 would unnecessarily clutter the figure. Or does the reviewer refer to his comment on Section 4.1.3, which is addressed below?
- "Title: Change in the title the part dealing with rethinking Australian stable continental region Earthquakes": Title text after the colon modified to more concisely introduce the paper content: "new insight into Australian stable continental region earthquakes"

- "For an international journal it would be more appropriate an introduction presenting data on moderate magnitude seismic sequences, their associated surface deformation and surface breaks (length)". We have rewritten the Introduction to better set the scope of the manuscript. We retain our focus on the stable continental region (SCR) setting, but have presented our study in the framework of global SCR earthquakes that precede the Lake Muir sequence, many of which have been imaged by InSAR. Table 1 has been expanded considerably to present data relating to these earthquakes.

- "Is it possible to better explain the meaning of grain in the landscape or add a reference for it?". We mean grain in the same way that trees have grain; a preferential alignment of constituent elements. To clarify, the word 'grain' in the sentence has been replaced with "an alignment of valleys and ridges".

- "At lines 285-295 the Authors infer fault geometry by aftershock distribution… To me this dataset is not enough to depict fault geometry". As a preamble to addressing this reviewer concern we note that in response to reviewer RC1 misunderstanding our aftershock deployment geometry we have added text to section 2.3 and to the caption of Figure 2 noting that the rapid deployment kits are prefixed by LM on Figure 2. We have also stated the uncertainties associated with the aftershock relocations in the text of Section 3.4, which was not made clear before. The mean location uncertainties in the relocated dataset were calculated to be 63 m, 116 m, and 228 m in the east, north and up directions, respectively. It is true that the computed aftershock distribution does not define a neat rupture plane, nor are relocated earthquakes so numerous, as may be the case in the plate margin examples presented by the reviewer. However, given the tight uncertainties we contend that the scatter is real, and the main concentration of aftershocks 'in general' occupy a volume defined by positive coulomb stress changes. This is especially the case when the Coulomb stress changes are resolved for optimal strike slip receiver faults. A new part to Figure 8 has been crafted to show this relationship. As the reviewer notes, this relationship remains less than 100%. However, we contend this stems in part from the 'real' scatter, and perhaps also to fluid migration of the sort described by Townend & Zoback (2000). We have tightened the text of Section 3.4, and the caption of Figure 8, to improve the clarity of communication of our observations.

- Section 3.5. "Relationship between moment magnitude and surface rupture length amongst Australian cratonic earthquakes. I suggest removing this paragraph". As this section was of concern to both reviewers, we have removed it. Instead, we focus on the utility of the InSAR data for deducing source parameters of each of the mainshock ruptures.

- Section 4.1.3. Co-location of thrust and strike-slip events: This is a quite big speculation since the resolution of the data do not allow for this, or data are not well presented to convince the reader about this. Provide an integrated picture to support the co-location. We must conclude that there are deficiencies in our presentation of the data as the evidence from the INSAR images is compelling and incontrovertible for an overlap of surface deformation envelopes resulting from the two largest events. This is explicitly stated in the text and is shown in Figure 6, where the surface trace of the strike-slip fault rupture relating to the November event (panels c and d) has been superposed onto the surface deformation pattern of the September event (panels a and b). We agree that there is significant uncertainty in relating the main shocks and aftershock distribution to geological structures. As mentioned in our general comments, we have tightened the uncertainties associated with the locations for the three largest events, and better communicated the uncertainties associated with the aftershock relocations. We contend that this allows us a firmer basis to explore the relationship between the rupture planes fro the larger events, and the aftershock locations.

- Section 4.1.3. "The sentence starting at line 443… In general, the volume in which aftershocks are located corresponds to a volume of positive Coulomb stress change resulting from the main shock (Figure 8)", is not 100% consistent with aftershock distribution… I suggest to significantly reduce this part and incorporate it in the discussion on the seismic sequence". As mentioned above, we have tightened the text in terms of communicating the uncertainties

associated with the hypocentral locations. Further, we have provided a potential explanation in the text for the hypocentres that do not occur in the volume of positive Coloumb stress. We contend that this justifies retaining the section.

- Section 4.1.4 "Mechanisms for strain localisation in Stable Continental Region (SCR) crust... This paragraph is not strongly related to the data presented in the manuscript but mainly based on literature. I suggest removing this part." This section removed as suggested.
- Section 4.2 "One-off ruptures from moderate to large magnitude earthquakes in the cratonic regions of Australia... I suggest removing this part." The section has been sharpened to emphasise that there is no evidence for prior rupture on the Lake Muir faults, and this is typical of Precambrian non-extended SCR crust worldwide, as presented in the reworded introduction. In addition, a section has been added prior (a new section 4.2) that discusses the stress triggering aspects of the new data, as recommended by reviewer RC1. The new sections 4.2 and 4.3 now focus on relating characteristics evident in the Lake Muir Ruptures with ruptures in analogous crust worldwide.
- Section 4.3. "Migration of the locus of moment release in the Southwest Seismic Zone… I suggest deleting it" This section has been deleted as suggested.
- Section 4.4. Deleted as suggested.

Reply to specific comments made by RC1:

- Section 2.3. "Rapid Deployment aftershock kits… Regarding the seismic station deployment, the closest station has been located at least at 24 km far from the epicentral area of both earthquakes... large uncertainties might afflict the location of small magnitude earthquakes occurred during the swarms". The reviewer misunderstood our communication of the experimental design. The nearest *permanent* network station is 24 km from the epicentral area. This is explicitly stated in the text. The five rapid deployment aftershock kits range in distance from right on top of the first main shock (LM01), to 42 km distant (LM05). We have added text to Section 2.3, and the caption of Figure 2, to make clear that the black triangles with labels prefixed by 'LM' on Figure 2 are the temporary stations. We understand that the network, comprising permanent and temporary stations, is not as dense as might be achieved in regions where there is a higher perception of earthquake hazard (i.e. non-SCR), but are satisfied that our uncertainties (better than 300 m for aftershocks) is suitable to make our conclusions. Further, relocation of the three largest events in the revised manuscript places us in a much better position to relate these events to structure and to the aftershocks.
- Section 3.5. "In my opinion this latter argument [new relationships between moment magnitude and surface rupture length] deserves a separate paper. The paper should be more focused to the expected results introduced by the title." This section has been removed as suggested by both reviewers. The wording has been changed throughout to celebrate the ability of the InSAR data to detect and characterise surface ruptures in remote areas, such as SCR crust.
- Line 445-448. "They don't clearly explain if they consider the November earthquake induced by a dynamic triggering of the September event …I think the clarification of this point could be an important statement for future and more addressed studies, for instance the fault ruptures interaction or the dynamic triggering between two or more seismic sources.' Excellent point! The text in Section 4.1.3 has been modified to reflect the fact that we do indeed think that the second M5 event was triggered. We have added a new section to the discussion (new section 4.2) that examines patterns common to the Lake Muir sequence and other Precambrian SCR sequences. Excitingly, we find evidence elsewhere that the triggered events can have different failure mechanisms.
- "in Figure 6a there is an east-west oriented fringe interruption at latitude/longitude 6190000/4790000? How the Authors interpret it?" The text of Section 3.2 states "Coherence is also partly lost beneath an approximately 2 km wide (N-S) easterly trending band of pine

forest (see Figures 3 and 6a for location)". The north-south extent of this forest is clearly marked with an arrow labelled "pine forest" on both Figures 3 and 6a. We have added the following text to the caption of Figure 6: The north-south extent of an easterly trending band of pine forest associated with degradation of coherence is indicated with a white arrow in part (a).

- "For a "not Australian" reader is a bit hard to follow the text with the lack of a clear tectonic map containing the bright place names tags." Figure 1 *is* a clear tectonic map as far as stable continental region crust may be divided with respect to seismogenic potential (e.g. Johnston et al 1994; Clark et al. 2012). In the revised manuscript, we have either included all Australian place names mentioned in the text in Figure 1, or written explicit locations into the text itself.

**References cited**

Townend, J., and Zoback, M. D.: How faulting keeps the crust strong, Geology, 28, 399-402, 2000.

**Surface deformation relating to  the 2018 Lake Muir earthquake sequence, southwest Western Australia:  new insight into  stable continental region earthquakes**

Dan J. Clark[1], Sarah Brennand[1], Gregory Brenn[1], Matthew C. Garthwaite[1], Jesse Dimech[1], Trevor I. Allen[1], Sean Standen[2]

[1] Positioning and Community Safety Division, Geoscience Australia, GPO Box 378 Canberra ACT, Australia
[2] The University of Western Australia, 35 Stirling Hwy, Crawley, Western Australia, Australia

*Correspondence to*: Dan J. Clark (dan.clark@ga.gov.au)

**Abstract.** A shallow $M_W$ 5.3 earthquake near Lake Muir in the stable continental region (SCR) crust of southwest Western Australia on the 16th of September 2018 was followed on the 8th of November by a proximal $M_W$ 5.2 event. Focal mechanisms produced for the events suggest reverse and strike-slip rupture, respectively. Field mapping, guided by Sentinel-1 InSAR data, reveal that the first event produced an approximately 3 km-long and up to 0.4-0.6 m high west-facing surface rupture, consistent with reverse slip on a moderately east-dipping fault. The InSAR data also shows that the surface scarp relates to a sub-surface rupture ~5 km long, bound at its north and southern extremities by  bedrock structures. The November event produced a surface deformation envelope that is spatially coincident with that of the September event, but did not result in discrete surface rupture. Almost nine hundred aftershocks were recorded by a temporary seismometer deployment. Hypocentre locations correlate poorly with the rupture plane of their respective main  shocks but correlate well with regions of increased coulomb stress. The spatial and temporal relationships between the $M_W$ > 5.0 events, and their aftershocks, reveals dependencies with  implications for how other less well documented SCR earthquake  sequences could be interpreted. Furthermore, the September $M_W$ 5.3 Lake Muir earthquake was the ninth event documented to have produced surface rupture in Australia in historical times. These nine ruptures are located exclusively in the Precambrian non-extended SCR rocks of central and western Australia, and none could have been identified and mapped using topographic signature prior to the historical event.  Consistent, though fragmentary, evidence exists from analogous regions worldwide. Our analysis of the Lake Muir earthquake sequence therefore provides constraint on models describing mechanisms for strain accumulation and localized release as earthquakes in non-extended SRC crust. ~~Modern geodetic and seismic monitoring tools are enabling the study of moderate sized earthquake sequences in unprecedented detail. Here we use a variety of methods to examine surface deformation caused by a sequence of earthquakes near Lake Muir in southwest Western Australia in 2018. A shallow MW 5.3 earthquake on the 16th of September 2018 was followed on the 8th of November 2018 by a MW 5.2 event in the same region. Focal mechanisms for the events suggest reverse and strike-slip rupture, respectively. Interferometric Synthetic Aperture Radar (InSAR) analysis of the events suggests that the~~

35  ~~confirms an east-dipping rupture plane for the first event, and shows a concentration located at the northern end of the rupture where the InSAR suggests greatest vertical displacement. The November event resulted from rupture on a northeast-trending strike-slip fault. UAV-derived digital terrain models (differenced with pre-event LiDAR) reveal a surface deformation envelope consistent with the InSAR for the first event, but could not discern deformation unique to the second event. New rupture length versus magnitude scaling relationships developed for non-extended cratonic regions as part of this study allow~~

40

The works published in this journal are distributed under the Creative Commons Attribution 4.0 License. The author's
45  copyright for this publication is transferred to the Commonwealth of Australia. The Creative Commons Attribution 4.0 License and the Commonwealth of Australia are interoperable and do not conflict with, reduce or limit each other.

© Commonwealth of Australia (Geoscience Australia) 2019.

[revised manuscript text omitted]

**Mainshock relocation**

The absolute location of the September Mw 5.3 Lake Muir mainshock determined by Geoscience Australia is not well
130 constrained by seismic stations of the Australian National Seismic Network. Furthermore, rapid deployment kit LM01 was

offline during the November mainshock, which added considerably to the location uncertainty. It was also not possible to

include the mainshocks with the HypoDD relative location of the aftershocks, as the aftershocks were mostly recorded by the aftershock kits, and there are few common local stations linking the aftershocks with the mainshocks. However, two factors make it possible to improve the locations for these largest events. 1) The three largest events (see main text, Table 2), were recorded on at least 19 common regional and teleseismic stations.  2) The October aftershock is well constrained because all of the rapid deployment kits were online, and the event occurred within a kilometre of station LM01.

It is possible in this case to do a three-event relative location, and then "anchor" the group to a known location (i.e. the October aftershock) to produce an improved absolute location of the two largest events. To do this we adapt the method of Fisk (2002), which uses a combination of manual waveform alignment and the Joint Hypocenter Determination (JHD) technique on regional and teleseismic phases to obtain an accurate relative location, and then anchors at least one event to a known surface location (in their case, satellite images of surface displacement).  While this method was developed for the high-precision location of nuclear weapons with repeatable waveforms, the flexibility provided by manual waveform alignment makes it possible to use in situations even where the waveforms do not perfectly match.  Our application is as follows:

1. Each earthquake is located using a suitable earthquake location program, using travel times read on the following Australian and IMS stations. AU: KDU, KLBR, BBOO, CMSA, KNRA, MEEK, MORW, MTKN, MTN, MUN, NAPP, QIS, STKA; IMS: CM31, KURBB, NWAO, QSPA.  The inclusion of QSPA in Antarctica is particularly important as it is the only station to the south of the event, and is needed to reduce the azimuthal gap.  In principle, any standard earthquake location algorithm or approach which is capable of handling regional and teleseismic phases can be used for the Fisk (2002) relative location strategy. We solve the seismic location problem using the "neighbourhood algorithm" approach (Sambridge and Kennett, 2001) and the AK135 travel times (Kennett et al., 1995), but we could have adapted a formal program such as LocSAT (Bratt and Nagy, 1991) or NonLinLoc (Lomax, 2008) to do the same.

2. Waveforms for each event at each station are manually aligned using the GeoTool program, and the P phases are re-picked (Figure S3).  In practice, this seems to improve the relative accuracy of the phase pick by about an order of magnitude or more. Like with the previous step, this outcome could also have been accomplished using another waveform analysis program such as SAC (Goldstein et al., 2003).

3. The average travel time residual of all three events is subtracted from the travel time of each event on a station-by-station basis, and the location is calculated again.  We fixed the earthquake depth at 2 km as the aftershock cloud begins at this depth, but we note that changing the depth between 0-10 km has little effect on the surface location as the nearest stations are relatively distant.  This process is iterated as many times as necessary, and we find that the residuals converge after just three iterations. This method is equivalent to the JHD approximation technique described in Pujol (2000), but any of the other JHD techniques mentioned there would have been appropriate as well.

4. The three events, which are now accurately located in a relative sense, are shifted so that the relatively located October $M_L$ 4.6 event overlies the October event location calculated using the aftershock kits.  This yields precise absolute locations of the September and November mainshocks.

5. Errors are calculated assuming that the precision of the travel time picks is within 4 samples. We use a modified Gaussian distribution with an L1 norm. This step is optional and is not intended to be extremely accurate. Rather, it is more indicative of the stability of the solution in terms of azimuthal coverage. There is ongoing debate regarding the appropriateness of error bars in relative locations, since a single poor phase pick can throw the solution out by orders of magnitude. It is therefore more important to ensure the data quality.

[Figure]

**Figure S3.** An example of a manual alignment of Lake Muir waveforms in GeoTool on the vertical component of station QIS. Top to bottom: September, November and October events respectively. "X" picks represent old phase picks. "P" picks are the new picks made after alignment, which is one pick made across all three events at the same time. The events were manually aligned by dragging them so that key features (e.g. A, B, and C) are aligned between them. These waveforms have been bandpass filtered 0.4-1.8 Hz (3rd order). While the P to X changes look small, the largest correction is 0.52 seconds. This makes a significant difference when calculating a relative location.

**Coulomb Stress modelling**

When slip occurs on a fault (the 'source' fault) stress is imparted to the surrounding crust and faults ('receiver' faults). In Coulomb stress modelling (e.g. Lin and Stein, 2004; Toda et al., 2005; Toda et al., 2011), fault displacements in the elastic half-space are used to calculate a 3D strain field, which is multiplied by the elastic stiffness to derive stress changes. Stress changes might be used to understand the distribution of aftershocks resulting from an event, or the static stress changes caused by displacement on a 'source' fault can be resolved onto 'receiver' faults to investigate whether they are promoted towards failure. The shear stress increase or decrease is dependent on the position, geometry, and slip of the source fault and on the position and geometry of the receiver fault, including its rake. The normal stress change (clamping or unclamping) is independent of the receiver fault rake (Toda et al., 2011).

Toda et al. (2011) use the Coulomb failure criterion, $\Delta\sigma_f = \Delta\tau_s + \mu' \Delta\sigma_n$, in which failure is hypothesized to be promoted when the Coulomb stress change is positive. Here, $\Delta\sigma f$ is the change in failure stress on the receiver fault caused by slip on the source fault(s), $\Delta\tau s$ is the change in shear stress (reckoned positive when sheared in the direction of fault slip), $\Delta\sigma n$ is change in normal stress (positive if the fault is unclamped), and $\mu'$ is the effective coefficient of friction on the fault.

195

The source fault was parametrised as follows: length = 5 km, strike = 5°, dip = 50°, dip slip = 0.6 m, strike slip = 0.0 m, rake = 90 °, top of rupture = 0 km, bottom of rupture = 0.9 km, co-efficient of friction ($\mu'$) = 0.6 (Figures 8 and S4). The preferred parametrisation for the  strike-slip receiver fault (i.e. the November rupture plane) was determined through simple forward modelling using a finite rectangular elastic dislocation model (Okada, 1985) (https://earthquakes.aranzgeo.com/model/generic1). The Parameters that were found to best match the surface expression are as follows: length = 4.1 km, strike = 233°, dip = 88°, dip slip = 0.0 m, strike slip = 0.0 m, rake = 0-°, top of rupture = 1.0 km, bottom of rupture = 2.5. km (Figure S5). This plane is shown in section on Figure S4b.

The Coulomb stress change consequent of the Lake Muir September $M_W$ 5.3 reverse fault rupture was resolved for  optimally oriented  strike-slip events parallel to the plane of the November $M_W$ 5.2 strike-slip rupture (Figure S4). This calculation shows that the plane of the November rupture was brought closer to failure over approximately half of its area, though only slightly so in the hypocentral region.

[Figure]

[Figure]

|210    **Figure S4: Coulomb stress change resulting from the September event resolved for dextral strike-slip events onto the plane of the November rupture. Stress increase values are shown in bars. The location and uncertainty ellipse of the hypocentre for the November event is shown.**

[Figure]

[Figure]

**Figure S5: Simple forward modelling of InSAR surface deformation envelope for the Lake Muir November mainshock. (a) InSAR phase image (recoloured from Figure 6C to enhance fringes). Fringes used for matching are delineated with white dashes. Preferred fault trace shown with a blue line ending in lack dots. (b) Forward model showing match to fringes (white dashed lines in part (a) are reproduced as black lines).**

215